neuroscience/behaviour/biomechanics

walking, gait, posture, vision, dynamics, stability

# Gazing down increases standing and walking postural steadiness

Yogev Koren[1], Rotem Mairon[2], Ilay Sofer[1], Yisrael Parmet[3], Ohad Ben-Shahar[2] and Simona Bar-Haim[1]

[1]Physical Therapy Department, [2]Computer Science Department, and [3]Industrial Engineering and Management Department, Ben-Gurion University of the Negev, Beer-Sheva, Israel

YK, 0000-0002-3287-279X; RM, 0000-0002-5756-2464; YP, 0000-0002-2071-7338; OB-S, 0000-0001-5346-152X; SB-H, 0000-0002-4368-4292

When walking on an uneven surface or complex terrain, humans tend to gaze downward. This behaviour is usually interpreted as an attempt to acquire useful information to guide locomotion. Visual information, however, is not used exclusively for guiding locomotion; it is also useful for postural control. Both locomotive and postural control have been shown to be sensitive to the visual flow arising from the respective motion of the individual and the three-dimensional environment. This flow changes when a person gazes downward and may present information that is more appropriate for postural control. To investigate whether downward gazing can be used for postural control, rather than exclusively for guiding locomotion, we quantified the dynamics of standing and walking posture in healthy adults, under several visual conditions. Through these experiments we were able to demonstrate that gazing downward, just a few steps ahead, resulted in a steadier standing and walking posture. These experiments indicate that gazing downward may serve more than one purpose and provide sufficient evidence of the possible interplay between the visual information used for guiding locomotion and that used for postural control. These findings contribute to our understanding of the control mechanism/s underlying gait and posture and have possible clinical implications.

**Author for correspondence:**
Yogev Koren
e-mail: yogevk@post.bgu.ac.il

## 1. Background

Visual information is important for guiding human locomotion [1,2]. When walking on uneven surface [3] or complex terrain [4], healthy individuals tend to gaze downward, presumably to reduce the surface's uncertainty [5] by identifying individual

footholds and use this information, in feed-forward [6] and/or online [7–9], to guide the leg's trajectory. Given a well-established relation between gaze behaviour and footholds in precise stepping paradigms (e.g. [10,11]), and the increased stepping accuracy when foveal information [9] is available during swing phase [8] in such paradigms, this perspective is only logical. Evidence from similar paradigms, such as obstacle negotiation [12,13] and stair negotiation [14], extend this perspective to other situations where leg trajectory's accuracy is important.

It has been suggested that the extent to which an individual fixates on a foothold depends on the perceived challenge to walking stability [11]; or more generally, fixation time depends on the perceived relevancy to the task [5]. Recent reports [15,16] support such assumption, as they indicate that anxiety (fear of falling) also leads to downward gazing (DWG). Taken together, this evidence suggests that DWG, when a person is walking on uneven or complex terrain, is an attempt to consciously control each individual step, triggered when walking stability is perceived as compromised.

Visual information is not used exclusively for guiding locomotion; it is also important for postural control (e.g. [17–20]). This can be easily shown by measuring postural sway with and without visual information (i.e. with eyes closed), and observing the difference (usually, the ratio is calculated, i.e. Romberg ratio). Previous reports suggest that the visual information used for guiding locomotion and that used for postural control, are both influenced by the visual structure of the environment [18,21–23]. This visual structure, along with its associated visual flow, naturally changes when gazing downward. Therefore, postural control may provide a complementary, and in some cases an alternative, explanation for DWG behaviour. In support of such a possibility, DWG has been reported to enhance postural steadiness for standing [24] and walking [25] stroke survivors.

To the best of our knowledge, excluding the above-mentioned report, no other attempts have been made to study how gazing downward affects postural control during human locomotion. Thus, we wanted to test the hypothesis that DWG can serve as a way to alter the visual flow in order to gain useful information for postural control.

Walking stability is naturally an important outcome of such investigation. Since this term has no widely acceptable definition in the literature [26], in this paper the term 'stability' will be used under the definition, 'the ability to perform without falling' (standing or walking) and the term 'steadiness' under the definition 'regularity and/or consistency of the motor output' (e.g. the magnitude of the variance of a given parameter). No preliminary assumptions regarding the relation between these two terms have been made.

In this paper, we explore the effect of DWG on the steadiness of healthy adults while standing and walking. The results of these experiments are then discussed in the context of the altered visual information during DWG.

Our hypothesis was that DWG will enhance postural steadiness during both standing and walking.

# 2. Results

## 2.1. Experiment 1

In this experiment, we wanted to investigate the effect of DWG on standing postural steadiness. We started with a standing paradigm as postural steadiness is easily defined as the motion of the centre of mass (COM) about the fixed base of support (BOS). This motion is commonly estimated from the change in ground reaction forces (GRFs) and the trajectory of the centre of pressure (COP). We recruited 15 healthy adults and evaluated their postural sway under five visual conditions: eyes closed (EC), downward gazing at their feet (DWGF), downward gazing 1 m ahead (DWG1), downward gazing 3 m ahead (DWG3) and forward gazing (FG) at eye level approximately 4 m ahead. Participants were instructed to stand as still as possible on a force plate in a standardized, narrow-base stance for 30 s. Each participant was tested five times under each condition in random order.

As outcome measures for postural steadiness, three parameters derived from the stabilogram diffusion analysis (SDA) were used—namely, the short-term diffusion coefficient of sway, in two dimensions ($D_{rs}$) and in one dimension, on the $X$-axis ($D_{xs}$) and on the $Y$-axis ($D_{ys}$), quantifying planar, medio-lateral (ML) and anterior–posterior (AP) sway, respectively [27]. Although the mechanism underlying SDA is controversial (e.g. [28]), it offers the possibility to investigate the dynamics of postural sway. To test whether DWG alters the directionality of sway, as indication that changes are due to altered visual flow (see *Discussion* section), we also calculated the difference in magnitude between ML and AP sway (i.e. sway range difference, SRD) in each condition (for a full description see the *Methods* section).

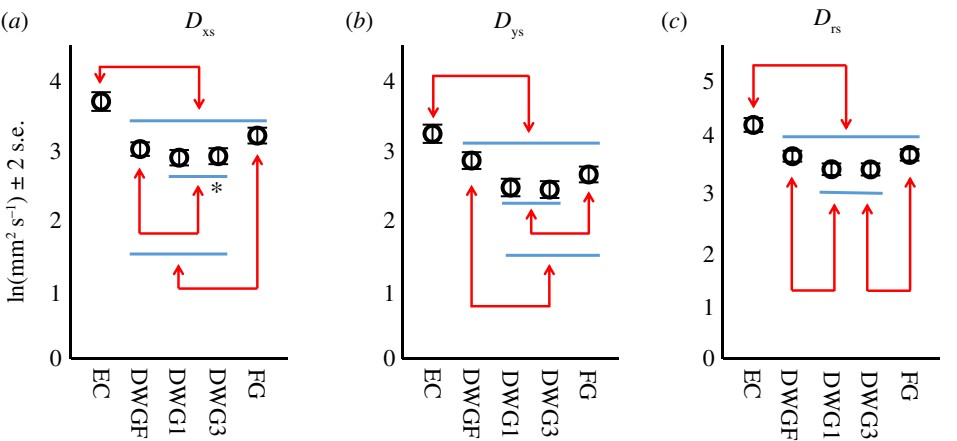

**Figure 1.** ($N = 15$)-pairwise comparison of mean short-term diffusion coefficient values (log-transformed) by condition for the parameters $D_{xs}$ (a), $D_{ys}$ (b) and $D_{rs}$ (c), quantifying ML, AP and planar sway, respectively. Mean values were calculated from 75 trials in each condition. Horizontal bars are grouping elements. Arrows indicate significant differences at the level of $\alpha < 0.05$. The [*] indicates $p = 0.055$.

The results of this experiment demonstrated a significant main effect for the visual condition in all SDA parameters ($F_{4,370} = 79.5$, $p < 0.001$; $F_{4,370} = 65.8$, $p < 0.001$ and $F_{4,370} = 91.2$, $p < 0.001$ for $D_{xs}$, $D_{ys}$ and $D_{rs}$, respectively). As expected, when visual information was unavailable (i.e. EC) sway values of all parameters were the greatest, signifying that visual information is important for postural control (figure 1). When visual information was available, mean sway values were the lowest, for all parameters, during the DWG1 and DWG3 conditions (for pairwise comparison see figure 1). The results of the directionality testing also revealed a main effect for the visual condition ($F_{4,370} = 5.61$, $p < 0.001$). Individual results revealed that mean SRD in all DWG conditions was not different from zero, as indicated by the 95% CI of the mean (DWGF −4.15–1.6; DWG1 −0.81–4.95; DWG3 −0.91–4.85), suggesting isotropic sway (i.e. ML and AP sway magnitudes were equivalent). For both EC and FG conditions, mean values were significantly greater than zero, as indicated by the 95% CI of the mean (EC 2.51–8.27; FG 0.63–6.39), indicating that the magnitude of ML sway was greater than that of AP sway.

## 2.2. Experiment 2

Next, we wanted to assess whether DWG will also affect walking postural steadiness, which was our main research question. To answer this question, we tested 30 participants under five visual walking conditions, four of which were similar to those in the standing postural testing (i.e. DWGF, DWG1, DWG3, FG). The fifth condition, which served as baseline in this experiment, was unrestricted (UR) in which participants were free to act as they please. For DWGF, participants were instructed to look where they assumed their next step is going to be. All tests were performed using a treadmill with an embedded force plate able to record vertical GRF and the trajectory of the COP of the whole body. Participants were instructed to walk, at their preferred velocity, for four minutes in each condition. Walking was continuous with a 10 s break between conditions, the order of which was random. The only instructions given were 'look continuously on the specific target of the condition', and 'try to maintain your position in the middle of the treadmill but without looking'.

As outcome measures, we quantified the steadiness of the COM motion about the BOS as estimated from the GRF time series; specifically, we computed the local divergence exponent ($\lambda^*$) for the vertical GRF (Fz), COPX and COPY time series, representing the vertical, ML and AP motions, respectively. The divergence exponent is a commonly used parameter thought to reflect the local steadiness of walking [29]. Like SDA, $\lambda^*$ also captures the dynamics of the motor output. As secondary outcome measures, we calculated the variability (as median absolute deviation, MAD) of the parameters step-width and step-time, which are commonly used to estimate gait steadiness.

First, we evaluated whether participants maintained their intended AP position on the treadmill during the walks and whether velocities were different between conditions (figure 2). Results of the position testing revealed a main effect for the condition ($F_{4,144} = 6.3$, $p < 0.001$). On average, AP position during the DWG1 condition significantly differed from the intended position (mean = 5.9 cm, 95% CI 2.4–9.4 cm), indicating an increase in the walker-to-target distance. Results of the velocity

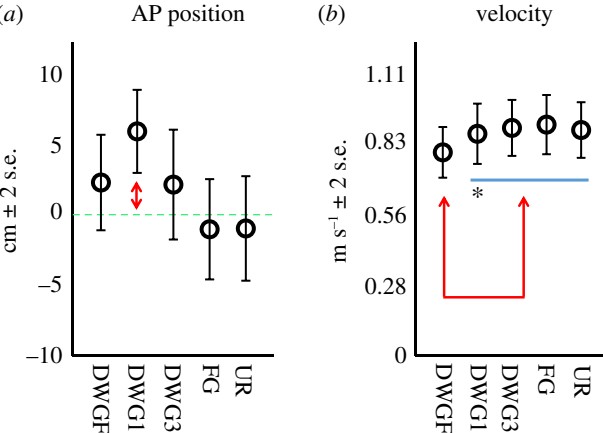

**Figure 2.** ($N = 30$)-mean AP centre of pressure position (*a*) relative to the intended position by condition, and pairwise comparison of the preferred walking velocities (*b*) by condition. The horizontal bar is a grouping element. Arrows indicate a significant difference at the level of $\alpha < 0.05$. The [*] indicates $p = 0.054$.

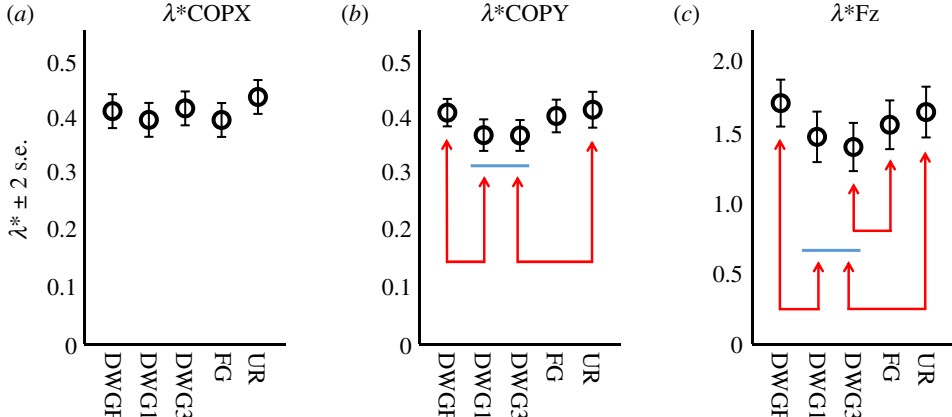

**Figure 3.** ($N = 30$)-pairwise comparison of mean divergence rate ($\lambda^*$) by gaze position of the COPX (*a*), COPY (*b*) and Fz (*c*) time series (ML, AP and vertical motions, respectively). For the last two, the predicted velocity-controlled values are presented. Horizontal bars are grouping elements. Arrows indicate significant differences at the level of $\alpha < 0.05$.

testing also revealed a main effect for the condition ($F_{4,145} = 2.6$, $p = 0.038$). Specifically, mean velocity during the DWGF condition was slower than velocities in the other conditions.

As for our main outcome measures, motion steadiness along the *X*- and *Y*-axes (ML and AP, respectively), as indicated by the parameters $\lambda^*$COPX and $\lambda^*$COPY, was unaffected by gaze position ($F_{4,145} = 1.5$, $p = 0.2$ and $F_{4,145} = 2.0$, $p = 0.1$, respectively), but steadiness of the vertical component, as indicated by the parameter $\lambda^*$Fz, was significantly affected by gaze position ($F_{4,145} = 5.4$, $p < 0.001$). To estimate whether $\lambda^*$ values were affected by walking velocity, as was previously reported [29,30], we tested these values against the self-selected velocity and found significant correlations with $\lambda^*$COPY and $\lambda^*$Fz values ($F_{1,148} = 12.0$, $p = 0.001$ and $F_{1,148} = 51.1$, $p < 0.001$, respectively). Therefore, we corrected for velocity by including this parameter as a fixed factor within the $\lambda^*$COPY and $\lambda^*$Fz statistical models. The results of the velocity-controlled models revealed that both parameters were significantly affected by gaze position ($F_{4,144} = 2.6$, $p = 0.04$ and $F_{4,144} = 4.2$, $p = 0.003$, respectively). In both models, mean values were the lowest during the DWG3 condition, but these were not significantly different from mean values during the DWG1 condition (see figure 3 for pairwise comparison).

To estimate walking steadiness, we also used step-time and step-width variability. Results of these models revealed a significant main effect of the condition for the former, but not the latter ($F_{4,144} = 3.13$, $p = 0.017$ and $F_{4,144} = 1.26$, $p = 0.29$, respectively), with the lowest mean value observed during the DWG3 condition. However, the former was significantly affected by walking velocity ($F_{1,148} = 131.1$, $p < 0.001$), and the velocity-controlled model revealed a non-significant effect of the condition (although a trend was noted) ($F_{4,144} = 2.32$, $p = 0.06$).

# 3. Discussion

The main purpose of this paper was to investigate the effect of DWG on postural steadiness. During walking, DWG is usually interpreted as an attempt at online or feed-forward control of stepping. By contrast, or maybe in conjunction, DWG has been shown to affect standing postural steadiness. Clark [31] reported that when toddlers were provided with external trunk support an adult-like walking pattern emerged. This was interpreted, from a dynamical systems perspective, as an indication that postural control is an important constraint on behaviour. Indeed, Kay & Warren [32] reported a complex relation between gait and posture that was sensitive to visual perturbations. Therefore, when considering how visual information is used to guide locomotion, one also has to consider the interplay with postural control.

## 3.1. Variables of the optic flow

From a visual information perspective, our results may be explained by the expected change in visual flow induced by the different gaze positions. For the purpose of this discussion we will consider three variables of the optical flow generated by relative motion: simple flow, which refers to planar or lamellar flow (parallel to the observer); motion parallax, which refers to differential flow due to parallax (most prominent when gaze is orthogonal to the motion direction); and radial expansion, which refers to a flow due to a differential rate of expansion (most prominent when gaze is in the direction of motion). For the latter, the term visual expansion seems more appropriate to the current discussion, since in most cases we consider the increase/decrease in the magnitude of a single, two-dimensional, object as its distance from the observer changes. Such scale changes seem sufficient to perceive motion [33]. It has been suggested that postural control is more sensitive to the last two variables [18–21] and that this sensitivity increases with flow density [34]. Moreover, in a stochastic three-dimensional environment, these two variables are congruent and will enable adequate postural control in any given direction. Since ML sway was reported to be greater than AP (i.e. anisotropy) during narrow-base stance [35,36], we expected that changes in visual flow would result in isotropy.

## 3.2. Experiment 1

Previous mechanistic investigations suggest that the combined biomechanical and vestibular effect of downward head inclination (i.e. neck flexion) decreases standing postural steadiness [37], but downward eye movement (i.e. eyes in head) was reported to increase postural steadiness [38,39]. These seemingly contradicting reports suggest that the need to control each individual step may come with some cost (or benefit) in terms of postural control. Our first experiment was designed to explore the effect of DWG on postural steadiness, when participants behave naturally (i.e. are free to use the head and neck, the eyes or both).

In the DWG1 and DWG3 conditions, the gaze is diagonal to the targets and therefore both near and distant features of the surface are contained within the visual field (around the target). This visual structure includes depth and will generate motion parallax for X-axis sway (ML) and visual expansion for Y-axis (AP) sway [19,21]. Thus, in both conditions, the visual flow generated by sway in any given direction is adequate for feedback control, and therefore equivalent and isotropic sway is expected, as was observed. During the DWGF condition, the gaze is perpendicular to the viewing target and provides a planar visual structure. Sway in this visual structure will generate simple flow in any given direction, which is less adequate for postural control. Therefore, sway in this condition is expected to be isotropic and increase in comparison to the DWG1 and DWG3 conditions, as was observed. FG, in the current study, also provides a planar visual structure as the target was placed on the back wall of the laboratory. Moreover, this was a white wall offering no or a minimal number of visual cues. Sway in this condition will generate visual expansion for motion along the Y-axis (AP) and simple flow along the X-axis (ML). Therefore, sway is expected to be anisotropic, with reduced sway along the Y-axis, as was observed. Y-axis sway in this condition is also expected to be reduced compared to the DWGF condition, which was observed, but equivalent with sway in the DWG1 and DWG3 conditions, which was not observed. X-axis sway in this condition is expected to be greater than in the DWG1 and DWG3 conditions, as was observed, but equivalent with sway in the DWGF condition, an outcome that was not observed. Both inconsistencies might be explained by the lower visual-cue density, provided by the wall.

### 3.2.1. Influences that are not dependent on optical flow

Not only ocular, but also extra-ocular information (i.e. proprioceptive information arising from the muscles controlling eye movement) affects postural sway [40]. This information is sensitive to gaze distance [19,38] and may provide an alternative explanation of our results. However, these previous mechanistic investigations had used very short gaze distances, which are irrelevant to walking. Moreover, under normal sway values, this information quickly decays to sub-threshold levels [40]. A comparison of the DWG1 and DWG3 conditions, in which the visual structure was similar, but gaze distance differed, supports this notion, as for all parameters tested no difference was found between them.

### 3.2.2. Summary

In this experiment, we did not try to investigate a specific mechanism/s, but to establish the overall effect of DWG to commonly reported distances during walking [3,4,11]. The results of this experiment were unambiguous, demonstrating that DWG to 1 and 3 m ahead significantly increased postural steadiness (but see contradicting results [24]). Although we suggest this effect was primarily derived from changes in visual information, other mechanisms/influences, such as biomechanical, proprioceptive and vestibular, are also quite possible.

## 3.3. Optical flow during walking

As in the standing experiment, the visual flow during walking was expected to differ among conditions. Specifically, we assumed that the visual flow would be similar to that created by standing sway on the horizontal plane. In addition, the visual flow created by vertical motion was also expected to differ among conditions. Vertical motion can create both motion parallax and visual expansion. The former is most prominent when gaze is parallel, and the latter when gaze is perpendicular, to the walking surface (assuming visual cues are within the visual field); thus, DWG just a few steps ahead (i.e. DWG1 and DWG3), as opposed to the DWGF and FG conditions, provides information that arises through both motion parallax and visual expansion.

Overground walking, as opposed to treadmill walking, also generates optic flow associated with continuous forward progression. This component was missing in our second experiment and will not be discussed, but see the *Limitations* section.

## 3.4. Experiment 2

For walking, we were able to identify only a single investigation of DWG's effect on postural steadiness [25]. According to this report, acceleration of the lumbar spine (at the level of L3 vertebra) was reduced in all three dimensions (i.e. vertical, ML and AP) when healthy older adults gazed down as opposed to forward, suggesting that DWG increases postural steadiness not only for standing but also for walking. Our second experiment was designed to explore this possibility.

### 3.4.1. Behavioural changes

The results of the second experiment were complex and not as straightforward as those of our standing experiment. First, we found that the preferred walking velocity in the DWGF condition was slower than in the other conditions (except for DWG1 in which a trend was noted, $p = 0.054$). We believe this was due to our instructions, which made participants attentive to each step (i.e. conscious control), thus, forcing additional task-related constraints. Next, we noted that participants increased gaze distance in the DWG1 condition by drifting backward on the treadmill, suggesting they were uncomfortable with the constraints of this condition. Gaze distance alone cannot explain this observation as gaze distance in the DWGF condition was shorter. We believe participants were trying (unconsciously) to optimize a certain parameter/s by increasing gaze distance. What this parameter is and whether it is related to optical flow or not are unclear at this time. However, it is not unlikely that participants considered the DWG1 target as a stepping target. It has been reported that humans most often fixate two steps ahead, when required to step precisely [11], and that such gaze behaviour seems optimal for feed-forward control when step precision is important [41]. Therefore, for individuals walking slowly, the DWG1 target might be at an optimal distance while for individuals walking faster this distance may be inadequate, i.e. too short.

## 3.4.2. Walking and postural steadiness

The main results of this experiment indicate that gaze position affected the steadiness of neither the motion along the $X$-axis (ML) nor along the $Y$-axis (AP), as indicated by the parameters $\lambda^*$COPX and $\lambda^*$COPY, respectively. However, after controlling for walking velocity, we found that $Y$-axis steadiness was significantly affected by gaze position. These results indicated that gazing down to 1 and 3 m ahead increased the steadiness of this motion, but by altering their walking velocity, participants were able to maintain a constant level of steadiness (however, see alternative explanation below). Further, we also used spatio-temporal parameters, which are more commonly used, and found that neither step-time variability nor step-width variability were affected by gaze position (although a trend was noted for the former). Thus, while both parameters relating to ML steadiness (step-width MAD and $\lambda^*$COPX) revealed no effect for the visual condition, the $\lambda^*$COPY did reveal such effect and the step-time MAD did not. This difference between the parameters relating to the AP steadiness may be the result of their sensitivity to the conditions, but we believe that the reason for the difference is that these parameters (step-time MAD and $\lambda^*$COPY) reflect different properties [42]. Specifically, the former quantifies postural steadiness and the latter quantifies the steadiness of gait (which are related); but even more importantly, for the former we eliminated any variability arising from gait cycle duration by unifying it across all gait cycles (see the *Methods* section), which is exactly the variability that the step-time MAD quantifies.

Interestingly, the parameters related to $X$-axis steadiness (i.e. $\lambda^*$COPX and step-width variability) did not correlate with walking velocity, an outcome that was unexpected. Since the treadmill used in this experiment was a split-belt type, it is possible that participants walked on a wider base than that to which they are accustomed. Therefore, the lack of effect on these parameters could be the result of our methodology, rather than a true lack of effect.

In addition to $Y$- and $X$-axes motions, walking includes a vertical component. To estimate its steadiness, we also calculated $\lambda^*$ for the GRF time series (i.e. $\lambda^*$Fz). Although this time series is not a trajectory per-se, it reflects the vertical acceleration (which is the second derivative of position) of the COM associated with gait cycle. Results of the $\lambda^*$Fz statistical model indicate that gaze position affected the steadiness of the vertical motion. Specifically, we found that DWG to 3 m ahead resulted in minimal $\lambda^*$Fz values, indicating maximal steadiness. Controlling for walking velocity within the $\lambda^*$Fz model resulted in minimal and equivalent values during the DWG1 and DWG3 conditions, as was observed in the $\lambda^*$COPY model. This was the most prominent effect in this experiment, indicating that DWG impacted primarily the vertical component.

As noted earlier, the results of this experiment were not as straightforward as those of the standing experiment and are, therefore, more difficult to interpret. Nevertheless, a previously published control model, in which gazing downward was suggested to serve in feed-forward when stepping onto a specific foothold (i.e. precise stepping) [41,43] may shed some light. Specifically, these authors suggest that during the single support phase, the body acts as an inverted pendulum, in which the COM rotates around the instantaneous BOS. They argue that once the execution of the step to the target commenced (after push-off), further control and the target itself were not required because the trajectory followed the pendular motion. These authors argue that such feed-forward control enables precise stepping, while exploiting the biomechanics of bipedal gait for energetic efficiency. This model suggests a close coupling between the vertical and forward motions, as they are determined by the radius of the pendulum. However, this inverted pendulum model assumes a fixed distance of the COM from the BOS, which is not the case for humans due to knee joint motion. Thus, the radius of the pendulum must be controlled throughout the single support phase to ensure the precision of the swinging leg. Our results suggest that this may be achieved through feedback control, by exploiting the visual flow created while gazing downward. If so, the effect observed for the $Y$-axis motion and that of the vertical motion might not be mutually exclusive, because both are influenced by the radius of the pendulum. Since we monitored the GRF and COP, and not the motion of the COM, the magnitudes of these components are not expected to be tightly coupled, but some resemblance is expected (as was observed).

This perspective not only demonstrates the interplay between the visual information used to guide locomotion and that used to control posture, but also provides a reasonable explanation as to why vertical steadiness decreased during the UR condition (in comparison to the DWG1 and DWG3 conditions)—that is, simply because step precision was unnecessary in this experiment. Namely, when free to act as they please, participants did not try to exploit the visual flow to achieve step precision simply because it was redundant, or in other words, steadiness was not perceived as necessary or beneficial to the goals of the task.

# 4. Conclusion

Overall, the findings from both our standing and treadmill-walking experiments indicate that DWG increases postural steadiness. These findings highlight the interplay between the visual information used to guide locomotion and that used for postural control, and should be considered when interpreting gaze behaviour. These results also demonstrate that steadiness is not necessarily beneficial, or otherwise desirable, unless dictated by the task's constraints or goals.

# 5. Clinical implications

As was mentioned earlier, a possible constraint on walking might be stability. It is possible that, under conditions of postural threat, humans resort to DWG (if steadiness contributes to their stability) as a means to avoid falling. Support for this possibility can be found in the literature. Ellmers *et al*. [15,16] reported that their participants gazed downward to task-irrelevant areas (but also relevant areas) when fear of falling was induced. Furthermore, in reports on gaze behaviour in elderly people and stroke survivors (populations at increased risk of falling [44–46]), elderly people gazed a shorter distance ahead than younger adults, while walking up and down stairs [14] and tilted their heads further down, while walking over a complex terrain [3], as well as on a flat surface [47]. As for stroke survivors, Aoki *et al*. [24] reported that 13 out of a sample of 15 stroke survivors were unable to walk 10 m, on a flat surface, without looking down, when instructed to do so.

# 6. Limitations

Several aspects of the current investigation might raise concerns for the interested reader. First, while walking on a treadmill, a person does not move forward, and thus the visual flow associated with such motion is absent. Although previous reports indicate that the presence or absence of this information had no effect on the sensitivity to postural-like, visual perturbations [18,21], a more recent report [23] suggests that this absence, in accordance with Weber's law, amplifies the perceived flow/signal, making it more appropriate for the central nervous system to use (but this is true only for the AP motion). Therefore, the ecological validity of our findings is unclear and should be further explored in more natural settings.

Another source of concern regarding our second experiment is the treadmill used in this investigation. This was a split-belt type treadmill, with roughly 2.5 cm between belts. In an attempt to avoid stepping between belts, our participants might have changed their step-width, as well as all associated parameters, and therefore we cannot be sure whether what we observed was a lack of effect or the consequence of our methodology.

Finally, although we suggest that the observed effects, in both experiments, are a consequence of altered visual flow (and provide a possible mechanism), other explanations are quite possible. Specifically, the downward head inclination may have biomechanical, vestibular and/or proprioceptive effects that might provide an alternative explanation. Therefore, mechanistic investigation of these effects should be used to confirm this perspective.

# 7. Methods

## 7.1. General

Participants in all experiments were healthy adults, recruited primarily from the university's undergraduate programmes. The advertisement was primarily through word of mouth and social media. In all cases participants were paid 11 USD in local currency. All experiments were approved, beforehand, by the ethics committee at Assaf Harofe Medical Center and conformed to the standards set by the Declaration of Helsinki (MOH_2018-02-14_002188). Participants were informed about the purpose and methods beforehand but information regarding the hypothesis was intentionally omitted. After they agreed to participate, written informed consent was obtained and demographic data collected. For descriptive statistics of these data, see table 1.

In all cases, statistical analysis was performed using linear mixed-effect models with participants as the *random* effect. For *post hoc* pairwise comparisons, paired-sample, two-sided *t*-tests were used. The

**Table 1.** Descriptive statistics.

|  | Experiment 1 | Experiment 2 |
| --- | --- | --- |
| male/female | 7/8 | 15/15 |
| mean age (range) (years) | 28 (20–41) | 27.6 (20–45) |
| mean weight (range) (kg) | 69.5 (51–94) | 70.5 (51–94) |
| mean height (range) (cm) | 170 (154–193) | 170.5 (153–193) |
| glasses yes/no | 9/6 | 15/15 |
| regular sports activity yes/no | 10/5 | 21/9 |

significance level was set to $\alpha < 0.05$, and the least significant difference (LSD) method was used to correct for multiple comparisons.

## 7.2. Experiment 1

Posturography of 15 adults, reporting no neurological or other condition affecting gait, was evaluated in this experiment. Participants were tested under five visual conditions: EC, DWG at their own feet/toes (DWGF), DWG1, DWG3 and FG at a target approximately 4.2 m ahead at eye level.

Participants were instructed to stand barefoot, as still as possible, on a Kistler 9286AA force platform (Kistler Instrument Corp., Winterthur, Switzerland) in a standardized stance, i.e. with their feet tight together and hands loosely hanging at their sides. Five 30 s quiet-standing trials in each of the five gaze positions were performed (with a total of 25 trials in random order). Raw data from the force plate were collected, at 100 Hz, using a data acquisition system consisting of a data acquisition box (Kistler A/D type 5691) and Bioware (v. 5.3.0.7) software.

Before each trial (i.e. a single 30 s stand) the force plate was calibrated with no weight (i.e. participants were instructed to step off the platform). Following the calibration, participants were instructed to stand on the platform and continuously look at one of the five targets. Locations for DWG1, DWG3 and FG were marked with coloured circles 20 cm in diameter. For the DWGF, participants were instructed to look at their own toes, while for the EC condition, no specific instructions were given besides 'close your eyes'.

The recordings were then processed by a dedicated MATLAB (MathWorks Inc. v. R2016b) script. First, the COP time series was low-passed using a second-order Butterworth filter with a cut-off frequency of 15 Hz. The script computes the short-term diffusion coefficient ($D_s$) of COP driven from SDA as described by Collins & De Luca [27]. Briefly, the diffusion coefficient is the rate at which the quadratic Euclidean distance between two COP positions increases as a function of the time interval between them. That is, for a given $\Delta t$, spanning $m$ data intervals and $N$ samples, planar displacement ($\Delta r^2$) is calculated as

$$D_{rs} = \left\langle \Delta r^2 \right\rangle = \frac{\sum_{i=1}^{N-m} (\Delta r_i)^2}{(N-m)}. \tag{7.1}$$

This calculation is repeated for every $\Delta t$, and the coefficient is calculated as the slope of the $\Delta$distance$^2$ by $\Delta t$ plot divided by two (i.e. $0.5 \times$ slope). In this experiment, we calculated three coefficients: single dimension on the $X$- ($D_{xs}$) and $Y$-axes ($D_{ys}$), and the planar coefficient ($D_{rs}$), all given in mm$^2$ s$^{-1}$. The short-term diffusion coefficients were reported to be reliable [27], more sensitive than COP-based summary statistics [48], and because they quantify the dynamic nature of steady-state stance, can be more informative regarding the underlying control mechanism. In general, smaller values indicate increased steadiness and are usually considered an indication of better postural control, while larger values are thought to be indicative of decreased steadiness due to impaired control. This perspective is derived from empirical evidence showing increase in sway when balance control is assumed to be reduced, such as when comparing sway with or without visual information [48,49], young and older adults [48], or elderly non-fallers and fallers [36,50].

To test for isotropy, the script also computed sway range along the $Y$- (AP) and $X$-axes (ML) (i.e. the distance between the extreme values on each axis) of each trial. The parameter SRD was then calculated as ML-AP. This parameter was computed as indication that the observed changes were due to change in the visual structure, under the assumption that in narrow-base stance ML sway is greater than AP [35]. In addition to sway range, the script also computes sway-velocity and area, which are more commonly used

and reported in such studies. Despite our initial concerns about the sensitivity of these parameters, the results of these 'traditional' parameters and those of the SDA-derived ones, were essentially identical. For the sake of brevity, the results of the traditional parameters are not reported.

For statistical analysis, we used all available measurements (i.e. every trial was represented as a data point). A total of 375 trials were obtained, two of which were excluded due to technical issues that arose during the experiments. These excluded values were replaced with the mean values of the participant in the specific condition. Since sway parameters' distribution significantly deviated from normal distribution (excluding SRD), we used a logarithmic transformation (denoted as Ln(*parameter*)). The transformed values were analysed using a linear mixed-effect model, with the condition as a *fixed* effect and subject as a *random* effect. Residuals of the models were evaluated for normal distribution.

## 7.3. Experiment 2

Thirty healthy adults were tested under five visual conditions: DWGF, DWG1, DWG3, FG and UR. For DWGF, participants were instructed to look where they assumed their next step is going to be, which for treadmill walking is about half a step forward. Participants were instructed to walk on a treadmill for 4 min, at their preferred walking velocity, while fixating on one of four visual cues (i.e. DWGF, DWG1, DWG3, FG) or simply walking without any instructions (i.e. UR). Compliance with these instructions were evaluated *post hoc* and found to be excellent in 62% of the trials, very good in 31%, and fair in 3%. In the remaining 4%, compliance could not be evaluated due to technical issues with the eye-tracking glasses. Eventually, all trials were included in the analysis (for details see electronic supplementary material, appendix).

Before testing, participants selected their preferred walking velocity in each of the conditions. We avoided using a single velocity under the assumption that the visual condition can affect this preference. Deviation from the preferred walking velocity was reported to decrease steadiness [51] and therefore can confound the results. For velocity selection, a random sequence of the conditions was computer-generated for each participant. Participants walked on the treadmill, starting at $0.056 \text{ m s}^{-1}$, while velocity was increased in increments of $0.028 \text{ m s}^{-1}$ at roughly 1 Hz. Participants were instructed to inform the examiner when the velocity of the treadmill was convenient and were allowed, if they so wished, to test velocities (increase or decrease velocity) before selection. This procedure was repeated for each condition according to the pre-generated random sequence. Once all velocities were obtained, testing commenced: a new random sequence was generated and participants walked, at their pre-selected velocity, in each condition according to this sequence. Between conditions, the treadmill was stopped for 10 s, during which, participants were instructed of their next condition. Each participant walked a total of 20 min.

Fixation targets for the DWG1, DWG3 and FG were marked as in the previous experiment. For DWGF, participants were instructed to fixate where they assumed their toes would be in the next step (i.e. one step ahead). For the UR condition, participants were instructed to behave as they felt comfortable. Since the treadmill is elevated from the laboratory's floor, a $3 \times 1.5 \times 0.2 \text{ m}$ wooden platform was custom built to create the illusion of continuity of the walking surface (with the targets for DWG1 and DWG3 attached to the platform). During all conditions, participants were instructed to try and walk in the middle of the treadmill to maintain a constant distance from the fixation targets. The Y-coordinates of this position were obtained beforehand, to control for any significant deviations during the experiment.

The treadmill used in this experiment was Mill (ForceLink, Culemborg, The Netherlands), a split-belt type treadmill with an embedded force plate, able to record vertical GRF and COP trajectory of the whole body. Data were collected continuously at 300 Hz throughout the experiment. Since the treadmill is equipped with a front holding bar and its motors are mounted in the front, all experiments were executed in the treadmill's reverse mode, thus, creating an imaginary, obstacle-free, walking path.

For processing, raw data from the treadmill's force plate were exported to MATLAB. Using a dedicated script, individual time series were first smoothed with a moving average of 10 points, after which individual heel-strikes were identified: a surge in the vertical force, exceeding body weight, that followed a rapid change in COPX trajectory, was identified from the GRF and COPX time series. The heel-strike was defined as the time point at which maximal inclination, along the surge, was identified. If the change in COPX was to the right, a right heel-strike followed and vice versa. From this series of heel-strikes the parameter *step-time* was computed as the time elapsed between consecutive heel-strikes. The parameter *step-width* was also computed and defined as the distance, in mm, between the X-coordinates of two consecutive heel-strikes.

Based on these computations we calculated centre and spread measures for each trial and parameter, excluding the first and last 10 steps of each trial. Following the work of Chau *et al.* [52] we evaluated the robustness of these measures by testing mean- and median-based measures. To make no assumptions we used the Wilcoxon signed-rank test. Results of these tests revealed significant differences ($p < 0.001$) between mean and median values of all parameters and also between standard deviations (s.d.) and the MAD of all parameters. Therefore, we decided to use the more robust estimates, i.e. median and MAD.

$$\mathrm{MAD}(X) = \mathrm{median}(|x_i - \mathrm{median}(x)|). \tag{7.2}$$

As primary outcome measures, we computed the *finite-time* maximal Lyapunov exponent (i.e. divergence exponent, $\lambda^*$) of the GRF (Fz), COPY and COPX time series. Lyapunov exponents ($\lambda$) quantifies the sensitivity of a dynamical system to initial conditions. An $n$-dimensional dynamical system can be characterized by the rate at which close trajectories diverge over time. Under certain conditions, this rate of divergence along a given coordinate converges to an exponential relation, which can be simply described as follows:

$$d(t) = d_0 e^{\lambda t}, \tag{7.3}$$

where $d$ represents the average separation as a function of time $t$, $d_0$ is the initial separation and $\lambda$ is the Lyapunov exponent of the system along this coordinate. Since the system is $n$-dimensional, it is characterized by an $n$-sized spectrum of lambdas. The maximal Lyapunov exponent ($\lambda_{\max}$) is simply the largest $\lambda$ in the spectrum. As $t$ grows larger, $\lambda_{\max}$ grows more and more dominant and the other exponents become negligible. When the evolution equation of the system is known, the Lyapunov spectrum is well defined in infinitesimal terms. However, estimating the Lyapunov spectrum of a hypothetical underlying system from a finite time series of measurements assumed to be produced thereby is by no means trivial. Various methods have been suggested for such estimation, but often it is only the maximal exponent that is of interest as a single indicator of the systems' sensitivity to initial conditions. Rosenstein *et al.* [53] proposed a relatively simple way of estimating just $\lambda_{\max}$ (denoted as $\lambda^*$) from a finite time series. This method has been widely used in various fields of research, including human locomotion [29]. Due to its simplicity, we adopted it too, although other methods [54] have been used as well.

To expose the Lyapunov spectrum and other dynamic system properties, a one-dimensional series usually has to be 'unfolded' or embedded in higher dimensions. This may be achieved by creating time-delayed copies of the original series (i.e. method of delays [55]). The data are then treated as multi-dimensional rather than one dimension.

For walking, basically, any time-dependent state variable/s may be used to characterize the dynamic properties of the system (i.e. the human body) using $\lambda^*$. Any nearby positions of this variable, in state-space, represent nearly the same state and any one may be regarded as a perturbation of the other [29]; thus, $\lambda^*$ may be used to quantify the rate at which these two positions diverge, representing the local dynamic steadiness (or more accurately, the unsteadiness) of gait.

To characterize the dynamics of walking in the present experiment, we computed $\lambda^*$ for the 5D space reconstructed from the Fz, COPY and COPX time series, using the middle 150 strides [56]. First, the individual unfiltered time series were time-normalized, using spline interpolation, to 200 points per stride so as to unify gait cycle and overall series lengths [56]. Next, the phase-space reconstruction interval was found by the mutual information first minimum method described by Fraser & Swinney [57]. Notably, values for the different state-variables (i.e. Fz, COPY and COPX) were very different. Using the 'false nearest neighbours' [58], we tested a few examples and found that a minimal, and therefore optimal, dimension of 5 resulted for all. Others have found this number to be optimal for gait-derived data [29,30]. We set the value of this parameter to 5 and refrained from computing it for each series separately.

Rosenstein's average log divergence by lag graph was computed using the *Tisean* package [59], freely available at https://www.pks.mpg.de/~tisean/. Our MATLAB program only extracted the slope of the initial linear trend of this graph and plotted it. Since this slope is very loosely defined in the original paper, we tried various linear approximations. From these plots a prominent difference between the state-variables was observed: the initial rapid increase ends after roughly 1–1.5 steps, for the parameter Fz, after 1.5–2 strides for the parameter COPX and even more for the parameter COPY. Moreover, significant fluctuations are easily observable throughout the plot. Therefore, approximation of the linear trend was greatly influenced by the time-period used. To overcome this difficulty, and for consistency,

we did not constrain the linear approximation by time (i.e. to either step or stride), as is common in the literature [26], but by magnitude. Namely, the approximation was computed over the 10–90% range of the maximal value observed in the time series. The slope of this linear approximation (corresponding to the short-term $\lambda^*$ usually used in the literature) was used for statistical analysis.

Processing resulted in 150 data points for each parameter. These were exported to SPSS for statistical analysis. Since we anticipated a correlation between velocity and all other parameters, we first evaluated these correlations. Results of this procedure revealed correlations between step-time MAD, $\lambda^*$COPY and $\lambda^*$Fz, and velocity, but not for the other parameters (i.e. step-width MAD and $\lambda^*$COPX). To control for the effect of velocity, we included walking velocity as a fixed effect in the statistical models of the appropriate parameters. Both the velocity uncontrolled and controlled models are reported. To test the effect of gaze position on walking steadiness we used mixed-effect models with the subject as the *random* effect and condition as the *fixed* effect. The models' residuals were then evaluated for normal distribution.

Ethics. All experiments were approved, beforehand, by the ethics committee at Assaf Harofe Medical Center and conformed to the standards set by the Declaration of Helsinki (MOH_2018-02-14_002188).
Data accessibility. The datasets supporting the findings reported in this paper can be found in the electronic supplementary materials. The SPSS (v. 26) code, used for statistical analysis of this data, is also in the electronic supplementary materials.
Authors' contribution. Y.K., I.S. and S.B.-H. were involved in all aspects of this work, including the conception and design of the experiments. R.M., Y.P. and O.B.-S. were involved in the analysis and interpretation of the data. All authors were involved in drafting and/or critically revising of this manuscript. All authors approved the final version of this manuscript and agree to be accountable for all aspects of this work.
Competing interests. We declare we have no competing interests.
Funding. This research was supported by the Helmsley Charitable Trust through the Agricultural, Biological and Cognitive Robotics Initiative and by the Marcus Endowment Fund, both at Ben-Gurion University of the Negev.

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
