## [Peer Review File · Royal Society Open Science]

Review History

RSOS-201556.R0 (Original submission)

Review form: Reviewer 1

Is the manuscript scientifically sound in its present form?

No

Are the interpretations and conclusions justified by the results?

No

Is the language acceptable?

Yes

Do you have any ethical concerns with this paper?

No

Have you any concerns about statistical analyses in this paper?

No

Recommendation?

Major revision is needed (please make suggestions in comments)

Comments to the Author(s)

The primary aim of this study was to determine whether gaze direction influences postural control during gait and stance. The results generally show that, in both stance and treadmill walking conditions, that downward directed gaze improves stability based on the observed reductions in the selected measures of stability. These results are relevant from both a basic science perspective (to understand how the balance system uses visual information) and a clinical perspective (implications for rehabilitation). This reviewer's major questions concern some of the methodological choices of outcome measures made by the authors in choosing parameters that quantify stability during stance and gait, and in the interpretation of visual motion cues available to subjects in the different test conditions.

Major comments:

1. For the Experiment 1 tests (stability during stance) why did the authors choose the short-term 'displacement' coefficient (Ds) from the Collins and DeLuca SDA analysis as their only outcome measure? One problem with quiet stance experiments is that there are too many quantitative measures to choose from. But why choose this one Ds parameter as opposed to others, such as a rms CoP sway and/or sway velocity measures, that are much more commonly used and for which readers are more likely to have an intuitive appreciation for? An SDA analysis generates 3 other parameters besides the Ds parameter so why choose Ds as opposed to one of the other 3? I note that the 1995 Collins paper shows quite large differences between young and old subjects in two of the other SDA parameters. The impression left by a selection of a single parameter is that the authors may have tried other parameters but only are showing us the one that produced results that were in agreement with their expectations. If this is true, then this is problematic. If this is not true and the authors had some a priori reason to select Ds as opposed to many others, then this should be discussed.
2. Somewhat of a similar question about the Experiment 2 analysis concerning the choice of parameter to characterize stability. The primary outcome was the Lyapunov exponent but also two other more conventional variability measures. The two conventional variability measures did not show significant gaze position effects but the Lyapunov measure did. There a fairly strong literature relating gait variability to clinically important factors like aging effects and falls prediction, so why no gaze-related difference with these measures but some with the Lyapunov measure? Perhaps the experimental results are constrained by a floor effect since the subjects were young and healthy.
3. This reviewer was also concerned about whether the visual scene was controlled enough to make strong conclusions about what aspect of vision was really contributing to the observed results. Specifically, the stimulus conditions only describe target sizes and locations. This tells us something about visual conditions near the center of the visual fields, but visual motion in the peripheral visual fields make important contributions to balance. For example, when viewing the more distant wall target in the FG condition, the authors describe this viewing as providing a planar visual structure (page 9 - I'm using page labels at the top of the page, line 49). available visual condition. But the subject's peripheral vision will also see the floor in front of them and probably side walls in the lab, and maybe there was other equipment in the lab that was visible. Also, the DWGF condition is complex since part of the field of view would include the subject's body which would move with the eyes/head and, therefore, a large portion of the visual field would not provide body-in-space motion information to control stance stability.
4. It's not clear what the expectations were for the UR condition in Experiment 2. Supplementary materials indicated that subjects gaze was monitored but it provides no summary

about the gaze in the UR condition. But it seems likely that different subjects would make different choices in controlling their gaze. At the end of the paragraph just before the Conclusions section (page 13), the authors attempt some explanation for results in the UR condition, but I can't say I understand this at all. The ending phrase is "simply because step precision was unnecessary in this experiment". Are you referring to the UR condition or to Experiment 2 in general?

5. Figure 1 and Experiment 1 analysis. The Drs measure is calculated by combining X and Y COP measurements so Drs is not independent of Dxs and Dys. Therefore it's a bit misleading to even bother showing and commenting on the Drs results separately since the implication, unless you think about it carefully, is that Drs results are somehow providing some independent confirmation for results seen in the other two measures, but that's not true. My suggestion would be to not include the Drs analysis.

6. All figures. A good general practice is to 'show the data'. All figures are only showing mean and two times standard errors. This gives a reader no feeling for the distribution of the data. Better would be to show the individual data points along side some summary representation (mean and SE or 2*SE).

Minor comments

1. Page 4, line 35. The general definition of steadiness in Experiment 1 is defined in terms of COM motion but the actual measures used in your analysis are derived from center of pressure (COP) motion which, for inverted pendulum mechanics, is related to the ankle torque exerted to control COM and is a function of COM displacement and COM acceleration.

2. Page 4, line 51 and page 16 line 14. The authors use the term "displacement coefficient" rather than the original "diffusion" coefficient. It seems unwise to redefine a well-established established term. Additionally, it's a slope measure and not a distance measure.

3. Figure 1 y-axis labels and page 16, line 33. Scientific units for seconds is 's' not 'sec'.

4. Page 6, line 29. Again, you are mentioning COM but your calculations are based on COP and Fz. Also, is referring to the BOS relevant for gait, as compared to stance, since the BOS is changing over time? Also, somewhere (probably Methods section) you should make clear that your COP measure is the whole body COP (I think it is) and not COP measure under each foot. Maybe your treadmill only has one force plate and that's what you get, but other systems can have separate force plates for each side.

5. Page 6, line 50. Does this 5.9 cm difference indicate that the subject was farther from the target or closer to the target?

6. Page 7, line 31. Should be 'values'.

7. Pages 9, 10. The discussion of isotropy and anisotropy is related to the visual conditions, but these X-Y differences will also be related to stance width (see some earlier work by Brian Day). I note that your data appears to show anisotropy in the EC condition which obviously would not be due to visual conditions.

8. Page 16, line 29. The use of the 'i' symbol on this line is confusing. Apparently it's meant to apply to X, Y, or R but 'i' is also used in the above equation.

9. Page 18, paragraph starting on line 18. This is not a very clear description. This explanation would benefit from have a methods figure that shows where your algorithm has picked off points from the COPX and GRF data indicating heel strike times.

10. Page 18, line 38. The Chau et al. reference is not in the reference list.

Review form: Reviewer 2

Is the manuscript scientifically sound in its present form?

Yes

Are the interpretations and conclusions justified by the results?

Yes

Is the language acceptable?

Yes

Do you have any ethical concerns with this paper?

No

Have you any concerns about statistical analyses in this paper?

No

Recommendation?

Major revision is needed (please make suggestions in comments)

Comments to the Author(s)

This manuscript investigates the contribution of gazing down towards postural control. Subjects stood and walked while gazing at different locations ranging from straight ahead to down at the feet. These two studies (standing, walking) found that gazing downward a few meters ahead resulted in more steady standing and walking, quantified by stabilogram diffusion analysis and Lyapunov exponents. These results have implications for the use of visual information for locomotion guidance and control.

Main concerns:

The manuscript is generally well-written and detailed. I have some main concerns that might provide additional clarity to the studies and to the implications of the results.

The manuscript was somewhat difficult to read through because of the extensive use of acronyms and X and Y (e.g. X-axis, Dxs, COPX). It was not clear to me initially which directions X and Y referred to. A brief explanation does exist in Results for Experiment 1 (Page 3), but it would be best if it were repeated in the Results section as well. Perhaps it would help the reader to change X and Y to ML and AP, respectively.

For Experiment 2, it is unclear why walking speed was not kept consistent throughout all the trials for each subject or among all subjects. This decision resulted in analyses that needed to account for velocity to show significance. Some explanation for the rationale behind letting subjects adjust walking speed between gaze conditions would be beneficial and any influence of allowing self-selected speed on the limitations of the experiment.

It is interesting that the UR (unrestricted) condition did not perform the best. The authors posited that perhaps step precision (which I interpreted as control) was unnecessary in this experiment. I am not sure about this explanation. Presumably, control is needed for all conditions to not, for example, fall off the treadmill. Do the authors mean that less control was needed for UR than for the DWG1 and DWG3 condition because it was more natural for the subjects?

One of the main conclusions is that downward gazing could be beneficial for postural control. For UR, did subjects prefer to adopt a downward gaze? Was it within one to three meters?

While I agree the explanation for gaze compliance should be in the appendix, the results section should give some indication that the subjects complied (and therefore the measures from these conditions are valid).

Minor comments:

Page 9 Line 24: "However, these previous mechanistic investigations had used gaze distances irrelevant to walking." Irrelevant in what way?

Page 10 Line 46: "We believe participants were trying (unconsciously) to optimize a certain parameter..." Please clarify what these certain parameters are.

Page 10 Line 52: Optimal for what? Related to energetics or stability or something else?

Page 11 Line 48: Do the authors mean "equivalent" or "similar", instead of "equivocal"?

Decision letter (RSOS-201556.R0)

Dear Mr Koren

The Editors assigned to your paper RSOS-201556 "Gazing Down Increases Standing and Walking Postural Steadiness" have now received comments from reviewers and would like you to revise the paper in accordance with the reviewer comments and any comments from the Editors. Please note this decision does not guarantee eventual acceptance.

Please submit your revised manuscript and required files (see below) no later than 21 days from today's (ie 11-Dec-2020) date. Note: the ScholarOne system will 'lock' if submission of the revision is attempted 21 or more days after the deadline. If you do not think you will be able to meet this deadline please contact the editorial office immediately.

on behalf of Dr Manoj Srinivasan (Associate Editor) and Kevin Padian (Subject Editor)
openscience@royalsociety.org

Editor Comments to Author:

Thanks for your submission. As you will see, the reviewers generally like the paper and accept your conclusions, which promise to be very useful. Their concerns are mainly with the mechanics of the paper, and so in your revisions please be sure to address each of these concerns. Best wishes,

Reviewer comments to Author:

Reviewer: 1

Comments to the Author(s)

The primary aim of this study was to determine whether gaze direction influences postural control during gait and stance. The results generally show that, in both stance and treadmill walking conditions, that downward directed gaze improves stability based on the observed reductions in the selected measures of stability. These results are relevant from both a basic science perspective (to understand how the balance system uses visual information) and a clinical perspective (implications for rehabilitation). This reviewer's major questions concern some of the methodological choices of outcome measures made by the authors in choosing parameters that quantify stability during stance and gait, and in the interpretation of visual motion cues available to subjects in the different test conditions.

Major comments:

1. For the Experiment 1 tests (stability during stance) why did the authors choose the short-term 'displacement' coefficient (Ds) from the Collins and DeLuca SDA analysis as their only outcome measure? One problem with quiet stance experiments is that there are too many quantitative measures to choose from. But why choose this one Ds parameter as opposed to others, such as a rms CoP sway and/or sway velocity measures, that are much more commonly used and for which readers are more likely to have an intuitive appreciation for? An SDA analysis generates 3 other parameters besides the Ds parameter so why choose Ds as opposed to one of the other 3? I note that the 1995 Collins paper shows quite large differences between young and old subjects in two of the other SDA parameters. The impression left by a selection of a single parameter is that the authors may have tried other parameters but only are showing us the one that produced results that were in agreement with their expectations. If this is true, then this is problematic. If this is not true and the authors had some a priori reason to select Ds as opposed to many others, then this should be discussed.
2. Somewhat of a similar question about the Experiment 2 analysis concerning the choice of parameter to characterize stability. The primary outcome was the Lyapunov exponent but also two other more conventional variability measures. The two conventional variability measures did not show significant gaze position effects but the Lyapunov measure did. There a fairly strong literature relating gait variability to clinically important factors like aging effects and falls prediction, so why no gaze-related difference with these measures but some with the Lyapunov measure? Perhaps the experimental results are constrained by a floor effect since the subjects were young and healthy.
3. This reviewer was also concerned about whether the visual scene was controlled enough to make strong conclusions about what aspect of vision was really contributing to the observed results. Specifically, the stimulus conditions only describe target sizes and locations. This tells us something about visual conditions near the center of the visual fields, but visual motion in the

peripheral visual fields make important contributions to balance. For example, when viewing the more distant wall target in the FG condition, the authors describe this viewing as providing a planar visual structure (page 9 – I'm using page labels at the top of the page, line 49). available visual condition. But the subject's peripheral vision will also see the floor in front of them and probably side walls in the lab, and maybe there was other equipment in the lab that was visible. Also, the DWGF condition is complex since part of the field of view would include the subject's body which would move with the eyes/head and, therefore, a large portion of the visual field would not provide body-in-space motion information to control stance stability.

4. It's not clear what the expectations were for the UR condition in Experiment 2. Supplementary materials indicated that subjects gaze was monitored but it provides no summary about the gaze in the UR condition. But it seems likely that different subjects would make different choices in controlling their gaze. At the end of the paragraph just before the Conclusions section (page 13), the authors attempt some explanation for results in the UR condition, but I can't say I understand this at all. The ending phrase is "simply because step precision was unnecessary in this experiment". Are you referring to the UR condition or to Experiment 2 in general?

5. Figure 1 and Experiment 1 analysis. The Drs measure is calculated by combining X and Y COP measurements so Drs is not independent of Dxs and Dys. Therefore it's a bit misleading to even bother showing and commenting on the Drs results separately since the implication, unless you think about it carefully, is that Drs results are somehow providing some independent confirmation for results seen in the other two measures, but that's not true. My suggestion would be to not include the Drs analysis.

6. All figures. A good general practice is to 'show the data'. All figures are only showing mean and two times standard errors. This gives a reader no feeling for the distribution of the data. Better would be to show the individual data points along side some summary representation (mean and SE or 2*SE).

Minor comments

1. Page 4, line 35. The general definition of steadiness in Experiment 1 is defined in terms of COM motion but the actual measures used in your analysis are derived from center of pressure (COP) motion which, for inverted pendulum mechanics, is related to the ankle torque exerted to control COM and is a function of COM displacement and COM acceleration.

2. Page 4, line 51 and page 16 line 14. The authors use the term "displacement coefficient" rather than the original "diffusion" coefficient. It seems unwise to redefine a well-established established term. Additionally, it's a slope measure and not a distance measure.

3. Figure 1 y-axis labels and page 16, line 33. Scientific units for seconds is 's' not 'sec'.

4. Page 6, line 29. Again, you are mentioning COM but your calculations are based on COP and Fz. Also, is referring to the BOS relevant for gait, as compared to stance, since the BOS is changing over time? Also, somewhere (probably Methods section) you should make clear that your COP measure is the whole body COP (I think it is) and not COP measure under each foot. Maybe your treadmill only has one force plate and that's what you get, but other systems can have separate force plates for each side.

5. Page 6, line 50. Does this 5.9 cm difference indicate that the subject was farther from the target or closer to the target?

6. Page 7, line 31. Should be 'values'.

7. Pages 9, 10. The discussion of isotropy and anisotropy is related to the visual conditions, but these X-Y differences will also be related to stance width (see some earlier work by Brian Day). I note that your data appears to show anisotropy in the EC condition which obviously would not be due to visual conditions.

8. Page 16, line 29. The use of the 'i' symbol on this line is confusing. Apparently it's meant of apply to X, Y, or R but 'i' is also used in the above equation.

9. Page 18, paragraph starting on line 18. This is not a very clear description. This explanation would benefit from have a methods figure that shows where your algorithm has picked off points from the COPX and GRF data indicating heel strike times.

10. Page 18, line 38. The Chau et al. reference is not in the reference list.

Reviewer: 2

Comments to the Author(s)

This manuscript investigates the contribution of gazing down towards postural control. Subjects stood and walked while gazing at different locations ranging from straight ahead to down at the feet. These two studies (standing, walking) found that gazing downward a few meters ahead resulted in more steady standing and walking, quantified by stabilogram diffusion analysis and Lyapunov exponents. These results have implications for the use of visual information for locomotion guidance and control.

Main concerns:

The manuscript is generally well-written and detailed. I have some main concerns that might provide additional clarity to the studies and to the implications of the results.

The manuscript was somewhat difficult to read through because of the extensive use of acronyms and X and Y (e.g. X-axis, Dxs, COPX). It was not clear to me initially which directions X and Y referred to. A brief explanation does exist in Results for Experiment 1 (Page 3), but it would be best if it were repeated in the Results section as well. Perhaps it would help the reader to change X and Y to ML and AP, respectively.

For Experiment 2, it is unclear why walking speed was not kept consistent throughout all the trials for each subject or among all subjects. This decision resulted in analyses that needed to account for velocity to show significance. Some explanation for the rationale behind letting subjects adjust walking speed between gaze conditions would be beneficial and any influence of allowing self-selected speed on the limitations of the experiment.

It is interesting that the UR (unrestricted) condition did not perform the best. The authors posited that perhaps step precision (which I interpreted as control) was unnecessary in this experiment. I am not sure about this explanation. Presumably, control is needed for all conditions to not, for example, fall off the treadmill. Do the authors mean that less control was needed for UR than for the DWG1 and DWG3 condition because it was more natural for the subjects?

One of the main conclusions is that downward gazing could be beneficial for postural control. For UR, did subjects prefer to adopt a downward gaze? Was it within one to three meters?

While I agree the explanation for gaze compliance should be in the appendix, the results section should give some indication that the subjects complied (and therefore the measures from these conditions are valid).

Minor comments:

Page 9 Line 24: "However, these previous mechanistic investigations had used gaze distances irrelevant to walking." Irrelevant in what way?

Page 10 Line 46: "We believe participants were trying (unconsciously) to optimize a certain parameter..." Please clarify what these certain parameters are.

Page 10 Line 52: Optimal for what? Related to energetics or stability or something else?

Page 11 Line 48: Do the authors mean "equivalent" or "similar", instead of "equivocal"?

===PREPARING YOUR MANUSCRIPT===

===PREPARING YOUR REVISION IN SCHOLARONE===

-- If you have uploaded ESM files, please ensure you follow the guidance at <https://royalsociety.org/journals/authors/author-guidelines/#supplementary-material> to include a suitable title and informative caption. An example of appropriate titling and captioning may be found at [https://figshare.com/articles/Table_S2_from_Is_there_a_trade-off_between_peak_performance_and_performance_breadth_across_temperatures_for_aerobic_sc ope_in_teleost_fishes_/3843624](https://figshare.com/articles/Table_S2_from_Is_there_a_trade-off_between_peak_performance_and_performance_breadth_across_temperatures_for_aerobic_scope_in_teleost_fishes_/3843624).

Author's Response to Decision Letter for (RSOS-201556.R0)

See Appendix A.

RSOS-201556.R1 (Revision)

Review form: Reviewer 1

Is the manuscript scientifically sound in its present form?

Yes

Are the interpretations and conclusions justified by the results?

Yes

Is the language acceptable?

Yes

Do you have any ethical concerns with this paper?

No

Have you any concerns about statistical analyses in this paper?

No

Recommendation?

Accept with minor revision (please list in comments)

Comments to the Author(s)

The primary aim of this study was to determine whether gaze direction influences postural control during gait and stance. The results generally show that, in both stance and treadmill walking conditions, that downward directed gaze improves stability based on the observed reductions in the selected measures of stability. These results are relevant from both a basic science perspective (to understand how the balance system uses visual information) and a clinical perspective (implications for rehabilitation). This reviewer was largely satisfied by the revisions and has only one major comment (that is easily fixed) and a few minor suggestions.

This reviewer did read the slightly revised version that was sent following the original submission of the revision, but the page and line numbers mentioned below refer to the original revision which did include line number labels.

Major comment

1. Page 14, line 37. The manuscript states that the diffusion coefficient was calculated as the slope of the $\Delta_{\text{distance}}^2$ by Δ_{t} plot. But the 1993 Collins and De Luca paper defines the diffusion coefficient as one half the slope (equation 1 and Figure 2b in Collins and De Luca; e.g. $\text{slope} = 2 \cdot D_{\text{rs}}$). If the authors actually are reporting the slope and not one half the slope, then Figure 1 and the description of the D calculation needs to be corrected. If the authors did the diffusion coefficient calculation correctly (according to Collins and De Luca) then the description of the calculation on page 14 needs to be corrected.

Minor comments

1. Page 3, line 45/46. This sentence mentions "a standardized, narrow-base stance". Exactly what the foot placement is for narrow-base stance should be defined in the Methods section. Were feet fully together or was there some space between the feet?
2. Page 5, line 46. A couple of spaces are needed in this parenthetical F statistic description.
3. Page 15, line 12. This sentence mentions how the results were "similar" to results from the SDA analysis when more traditional parameters were used. This is a bit of a weak statement. Can the authors make a stronger claim? Maybe something like "essentially identical such that overall conclusions were unchanged" (assuming this is true - if not true then some more details are necessary).
4. Page 15, lines 47-49. "ETG" is not defined yet. I see in the Appendix that this apparently stands for eye tracking device. It is not worth defining a new acronym here.
5. Page 15, first paragraph in section 6.3. Various percentages are given for how well subjects were able to comply with the gaze directions. It seems that there should be some correspondence between these percentages and those reported in the Appendix, but there doesn't seem to be.

6. Page 16, line 50. This description for identifying the heel strike point is not clear. First, as currently written the phrase “exceeding body weight” seems to be saying something about the “change in COPX trajectory” but apparently it should apply to the first part of the sentence. That is, “. . . a surge in GRF, exceeding body weight, followed the onset of a rapid change in COPX trajectory.” Then the following sentence is also not clear given the figure that the authors included in their response to the reviewers. As stated, the heel strike would be identified at the peak of the GRF since this is the point where there is a “maximal increase in force”. Maybe say “The heel-strike was defined as the point where the COPX begins its rapid change and the GRF begins its surge.” Or something similar.

7. I note that reviewer #2 wanted to see that ML and AP labeling was used rather than, or in addition to, the X and Y labels. This should also apply to the figure legends and, better yet, the headings above the panels in Figures 1 and 3 should have the ML and AP labels.

Review form: Reviewer 2

Is the manuscript scientifically sound in its present form?

Yes

Are the interpretations and conclusions justified by the results?

Yes

Is the language acceptable?

Yes

Do you have any ethical concerns with this paper?

No

Have you any concerns about statistical analyses in this paper?

No

Recommendation?

Accept with minor revision (please list in comments)

Comments to the Author(s)

The authors have thoroughly addressed my concerns, and the manuscript is improved. I have a few remaining minor comments.

The amended text about subject compliance states, "and found to be excellent in 62% of the trials, very good in 31%, and fair in 3%." I am unsure how the authors determined or how readers should interpret what is considered excellent, very good, or fair, and there does not seem to be an explanation in the Appendix either.

The authors noted that they could not evaluate whether downward gaze was adopted for the UR condition because gaze was only evaluated for subject compliance. It would be beneficial to include that in the manuscript, perhaps in the Appendix. A natural question leading from the results is whether the UR gaze was in the downward direction, and that note would let the readers know that it remains unknown.

Supplementary files containing data are in .sav and .sps, which I cannot open. I am not sure if supporting data is included with these SPSS files. The data files should be included in a format that is accessible by most software programs (e.g. in .csv or .txt).

Decision letter (RSOS-201556.R1)

Dear Mr Koren

On behalf of the Editors, we are pleased to inform you that your Manuscript RSOS-201556.R1 "Gazing Down Increases Standing and Walking Postural Steadiness" has been accepted for publication in Royal Society Open Science subject to minor revision in accordance with the referees' reports. Please find the referees' comments along with any feedback from the Editors below my signature.

Please submit your revised manuscript and required files (see below) no later than 7 days from today's (ie 26-Jan-2021) date. Note: the ScholarOne system will 'lock' if submission of the revision is attempted 7 or more days after the deadline. If you do not think you will be able to meet this deadline please contact the editorial office immediately.

on behalf of Dr Manoj Srinivasan (Associate Editor) and Kevin Padian (Subject Editor)
openscience@royalsociety.org

Editorial Comments:

We note that there were some issues regarding the figures within your submission. When submitting your finalised revision, please ensure to include your final figures in-text where you wish for them to be placed (along with the appropriate figure caption), and as separate figure files for our Production team.

Associate Editor Comments to Author (Dr Manoj Srinivasan):

Both reviewers agree that the authors have revised the article to mostly address their comments and suggestions. They have a few more minor remarks and one substantial one, which can still be addressed by edits to the text (to provide the description suggested by the reviewer). We look forward to a revised version.

Reviewer comments to Author:

Reviewer: 2

Comments to the Author(s)

The authors have thoroughly addressed my concerns, and the manuscript is improved. I have a few remaining minor comments.

The amended text about subject compliance states, "and found to be excellent in 62% of the trials, very good in 31%, and fair in 3%." I am unsure how the authors determined or how readers should interpret what is considered excellent, very good, or fair, and there does not seem to be an explanation in the Appendix either.

The authors noted that they could not evaluate whether downward gaze was adopted for the UR condition because gaze was only evaluated for subject compliance. It would be beneficial to include that in the manuscript, perhaps in the Appendix. A natural question leading from the results is whether the UR gaze was in the downward direction, and that note would let the readers know that it remains unknown.

Supplementary files containing data are in .sav and .sps, which I cannot open. I am not sure if supporting data is included with these SPSS files. The data files should be included in a format that is accessible by most software programs (e.g. in .csv or .txt).

Reviewer: 1

Comments to the Author(s)

The primary aim of this study was to determine whether gaze direction influences postural control during gait and stance. The results generally show that, in both stance and treadmill walking conditions, that downward directed gaze improves stability based on the observed reductions in the selected measures of stability. These results are relevant from both a basic science perspective (to understand how the balance system uses visual information) and a clinical perspective (implications for rehabilitation). This reviewer was largely satisfied by the revisions and has only one major comment (that is easily fixed) and a few minor suggestions.

This reviewer did read the slightly revised version that was sent following the original submission of the revision, but the page and line numbers mentioned below refer to the original revision which did include line number labels.

Major comment

1. Page 14, line 37. The manuscript states that the diffusion coefficient was calculated as the slope of the $\Delta_{\text{distance}}^2$ by Δ_{t} plot. But the 1993 Collins and De Luca paper defines the

diffusion coefficient as one half the slope (equation 1 and Figure 2b in Collins and De Luca; e.g. slope = $2 * D_{rs}$). If the authors actually are reporting the slope and not one half the slope, then Figure 1 and the description of the D calculation needs to be corrected. If the authors did the diffusion coefficient calculation correctly (according to Collins and De Luca) then the description of the calculation on page 14 needs to be corrected.

Minor comments

1. Page 3, line 45/46. This sentence mentions “a standardized, narrow-base stance”. Exactly what the foot placement is for narrow-base stance should be defined in the Methods section. Were feet fully together or was there some space between the feet?
2. Page 5, line 46. A couples of spaces are needed in this parenthetic F statistic description.
3. Page 15, line 12. This sentence mentions how the results were “similar” to results from the SDA analysis when more traditional parameters were used. This is a bit of a weak statement. Can the authors make a stronger claim? Maybe something like “essentially identical such that overall conclusions were unchanged” (assuming this is true – if not true then some more details are necessary).
4. Page 15, lines 47-49. “ETG” is not defined yet. I see in the Appendix that this apparently stands for eye tracking device. It is not worth defining a new acronym here.
5. Page 15, first paragraph in section 6.3. Various percentages are given for how well subjects were able to comply with the gaze directions. It seems that there should be some correspondence between these percentages and those reported in the Appendix, but there doesn't seem to be.
6. Page 16, line 50. This description for identifying the heel strike point is not clear. First, as currently written the phrase “exceeding body weight” seems to be saying something about the “change in COPX trajectory” but apparently it should apply to the first part of the sentence. That is, “. . . a surge in GRF, exceeding body weight, followed the onset of a rapid change in COPX trajectory.” Then the following sentence is also not clear given the figure that the authors included in their response to the reviewers. As stated, the heel strike would be identified at the peak of the GRF since this is the point where there is a “maximal increase in force”. Maybe say “The heel-strike was defined as the point were the COPX begins its rapid change and the GRF begins its surge.” Or something similar.
7. I note that reviewer #2 wanted to see that ML and AP labeling was used rather than, or in addition to, the X and Y labels. This should also apply to the figure legends and, better yet, the headings above the panels in Figures 1 and 3 should have the ML and AP labels.

===PREPARING YOUR MANUSCRIPT===

Your revised paper should include the changes requested by the referees and Editors of your manuscript. You should provide two versions of this manuscript and both versions must be provided in an editable format:
 one version identifying all the changes that have been made (for instance, in coloured highlight, in bold text, or tracked changes);
 a 'clean' version of the new manuscript that incorporates the changes made, but does not highlight them. This version will be used for typesetting.

===PREPARING YOUR REVISION IN SCHOLARONE===

-- Ensure that your data access statement meets the requirements at <https://royalsociety.org/journals/authors/author-guidelines/#data>. You should ensure that you cite the dataset in your reference list. If you have deposited data etc in the Dryad repository, please only include the 'For publication' link at this stage. You should remove the 'For review' link.

Author's Response to Decision Letter for (RSOS-201556.R1)

See Appendix B.

RSOS-201556.R2 (Revision)

Review form: Reviewer 1

Is the manuscript scientifically sound in its present form?

Yes

Are the interpretations and conclusions justified by the results?

Yes

Is the language acceptable?

Yes

Do you have any ethical concerns with this paper?

No

Have you any concerns about statistical analyses in this paper?

No

Recommendation?

Accept as is

Comments to the Author(s)

No additional modifications are requested.

Review form: Reviewer 2

Is the manuscript scientifically sound in its present form?

Yes

Are the interpretations and conclusions justified by the results?

Yes

Is the language acceptable?

Yes

Do you have any ethical concerns with this paper?

No

Have you any concerns about statistical analyses in this paper?

No

Recommendation?

Accept as is

Comments to the Author(s)

The authors have addressed my concerns. The data files are now in an accessible format.

Decision letter (RSOS-201556.R2)

Dear Mr Koren,

It is a pleasure to accept your manuscript entitled "Gazing Down Increases Standing and Walking Postural Steadiness" in its current form for publication in Royal Society Open Science.

on behalf of Dr Manoj Srinivasan (Associate Editor) and Kevin Padian (Subject Editor)
openscience@royalsociety.org

Reviewer comments to Author:
Reviewer: 2

Comments to the Author(s)
The authors have addressed my concerns. The data files are now in an accessible format.

Reviewer: 1

Comments to the Author(s)
No additional modifications are requested.

Appendix A

Editor Comments to Author:

Thanks for your submission. As you will see, the reviewers generally like the paper and accept your conclusions, which promise to be very useful. Their concerns are mainly with the mechanics of the paper, and so in your revisions please be sure to address each of these concerns. Best wishes,

Reviewer comments to Author:

Dear Editor and Reviewers,

We wish to thank you all for the time and effort invested in our manuscript. Below you can find our point-by-point response to your comments. In general, we found these comments extremely helpful. In just a few cases we left it up to the editor's discretion to decide if and how to make the appropriate revision. Although we do not object to any of the suggested revisions, in these cases we believe the revision will not add much.

Before addressing each individual comment, we wish to make a general statement regarding the first two comments (made by Reviewer 1) concerning the main outcome measures selected for our experiments. Specifically, this choice was made for two main reasons. First, both the diffusion coefficient and the divergence exponent quantify steadiness from a dynamic perspective. We believe that this approach captures the nature of the quantified behavior better than summary statistics. Second, both outcome measures use the ground reaction forces and the trajectory of the Center of Pressure (COP) as their input variables, thus creating a continuum between the two experiments. Although we agree with Reviewer 1 that the motion of the COP does not reflect solely the motion of the Center of Mass (COM) and agree that this motion is much smaller than that of the COM, it does reflect the forces applied to the surface— either to control its motion (as in planter flexion to decelerate the motion of the COM when it is moving forward) or to direct its motion (as in plantar flexion to move the COM backwards). Namely, both main outcome measures reflect the dynamic nature of postural control.

Reviewer: 1

Comments to the Author(s)

The primary aim of this study was to determine whether gaze direction influences postural control during gait and stance. The results generally show that, in both stance and treadmill walking conditions, that downward directed gaze improves stability based on the observed reductions in the selected measures of stability. These results are relevant from both a basic science perspective (to understand how the balance system uses visual information) and a clinical perspective (implications for rehabilitation). This reviewer's major questions concern some of the methodological choices of outcome measures made by the authors in choosing parameters that quantify stability during stance and gait, and in the interpretation of visual motion cues available to subjects in the different test conditions.

Major comments:

1. For the Experiment 1 tests (stability during stance) why did the authors choose the short-term 'displacement' coefficient (Ds) from the Collins and DeLuca SDA analysis as their only outcome measure? One problem with quiet stance experiments is that there are too many quantitative measures to choose from. But why choose this one Ds parameter as opposed to others, such as a rms CoP sway and/or sway velocity measures, that are much more commonly used and for which readers are more likely to have an intuitive appreciation for? An SDA analysis generates 3 other parameters besides the Ds parameter so why choose Ds as opposed to one of the other 3? I note that the 1995 Collins paper shows quite large differences between young and old subjects in two of the other SDA parameters. The impression left by a selection of a single parameter is that the authors may have tried other parameters but only are showing us the one that produced results that were in agreement with their expectations. If this is true, then this is problematic. If this is not true and the authors had some a priori reason to select Ds as opposed to many others, then this should be discussed.

Since its introduction in 1993 [1], Stabilogram-diffusion analysis (SDA) has been adopted by several research groups that have shown that SDA parameters are sensitive to vision [2], aging [2,3], Parkinson's disease [4], history of falls [5,6] and more. Further, SDA-based parameters were reported to be more sensitive to the effect of aging than summary statistics-based parameters [2]. Collins and De-Luca [1] reported that, among the SDA-based parameters, the short-term coefficients had the highest reliability, ranging from ICC=0.83-0.9. The scaling exponents of the short-term region were also reported to be reliable (ICC= 0.76-0.92), but they have rarely been reported in the literature, possibly because they are not as intuitive as the diffusion coefficients. The ICC values of the other parameters were much lower (when considering all 3 variables, i.e. two 1D and one 2D).

As noted by the reviewer (second comment), a floor-effect is quite possible in healthy adults. This is not just a hypothetical possibility in the current investigation, as Aoki et al. [7] conducted a similar investigation, using sway RMS and range parameters, and found no effect for downward gazing in healthy adults standing on two legs. They did report such effect for one-legged stance [7] and for stroke survivors [8]. With this in mind, we naturally looked at "traditional" parameters as well. Specifically, we looked at sway range, velocity and area; however, despite our initial concerns, the results of these parameters were similar to those of the SDA-based parameters (see an example in Figure 1 below). Because we believe that SDA-derived parameters are more informative, only the SDA parameters were reported for the sake of brevity and for continuum with the walking experiment. We have revised the manuscript to address the reviewer's concern (Section 6.2, paragraphs 6-7).

Figure 1: Sway velocity and the planar diffusion coefficient (Drs), \log_e transformed by the different visual conditions: eyes closed, downward gazing onto feet (DWGF), downward gazing one meter ahead (DWG1), downward gazing three meters ahead (DWG3), and forward gazing (FG). Note that this is raw data (i.e., summary statistics) and not the result of statistical analysis.

.2. Somewhat of a similar question about the Experiment 2 analysis concerning the choice of parameter to characterize stability. The primary outcome was the Lyapunov exponent but also two other more conventional variability measures. The two conventional variability measures did not show significant gaze position effects but the Lyapunov measure did. There a fairly strong literature relating gait variability to clinically important factors like aging effects and falls prediction, so why no gaze-related difference with these measures but some with the Lyapunov measure? Perhaps the experimental results are constrained by a floor effect since the subjects were young and healthy.

Although quantifying different properties, like SDA, the divergence exponent (also referred to as Local Dynamic Stability, LDS) quantifies steadiness from a dynamical perspective. LDS is relatively easy to understand and can be computed from any time series, including the ground-reaction forces (and COP) time series, to reflect walking postural steadiness, thereby creating a continuum with the standing experiment. Since Dingwell and Cusumano [9] first implemented the Rosenstein algorithm [10] to quantify walking steadiness and showed that LDS can distinguish between over-ground and treadmill walking and between neuropathic and healthy adults, LDS has gained popularity among researchers [11]. LDS has been shown to be sensitive to ACL ruptures [12], aging [13] and, specifically in fall-prone elderly [14], knee osteoarthritis [15], stroke [16], visual perturbations [17] and more. As noted by the reviewer, there is a large body of evidence that gait variability is related to clinically important factors, namely, increased variability. In this experiment, as opposed to the standing experiment, the results for the more “traditional” measures were different from those of the Lyapunov exponents. This difference was worth mentioning. That being said, a trend was noted for the step-time MAD ($p=0.06$) with minimal values observed during the DWG3 condition, results similar to those observed for the exponents of the vertical and AP motions. In Figure 2, below, we present the results of the step-time MAD and of the Fz time series (which is not presented in the manuscript). It can be easily seen that the results are quite similar,

but the step-time MAD results did not reach significance level. The reason might be a floor effect, as suggested by the reviewer, due to reduced sensitivity of the variability measures, but the results may also indicate that LDS and variability quantify different properties [18]. This second possibility is more likely, because we eliminated temporal variability from the LDS analysis by unifying gait-cycle duration before the analysis (see Methods, Section 6.3, paragraph 12). To address the reviewer's comment, we have added a short discussion about the difference observed between the variability measures and the divergency exponent (Section 3.4.2, paragraph 1).

Figure 2: Step-time variability (MAD) and the divergence exponent (Fz) for the visual conditions: downward gazing onto feet (DWGF), downward gazing one meter ahead (DWG1), three meters ahead (DWG3), forward gazing (FG) and unrestricted (UR). Note that this is raw data (i.e., summary statistics) and not the result of statistical analysis.

3. This reviewer was also concerned about whether the visual scene was controlled enough to make strong conclusions about what aspect of vision was really contributing to the observed results. Specifically, the stimulus conditions only describe target sizes and locations. This tells us something about visual conditions near the center of the visual fields, but visual motion in the peripheral visual fields make important contributions to balance. For example, when viewing the more distant wall target in the FG condition, the authors describe this viewing as providing a planar visual structure (page 9 – I'm using page labels at the top of the page, line 49). available visual condition. But the subject's peripheral vision will also see the floor in front of them and probably side walls in the lab, and maybe there was other equipment in the lab that was visible. Also, the DWGF condition is complex since part of the field of view would include the subject's body which would move with the eyes/head and, therefore, a large portion of the visual field would not provide body-in-space motion information to control stance stability.

We are unsure about this comment. The reviewer suggests that the specific visual structure of our laboratory might have affected our results and that in the DWGF condition we over-simplified our description of the visual field. While we agree with both statements, we do not quite understand why these issues would be significant. The objective of this investigation was to establish the effect of a downward gaze on

postural steadiness during standing and walking. While we suggest that the observed effect was due to the change in visual flow, we also stress that there are other possibilities and that an appropriate mechanistic investigation should be conducted in the future (see Section 3.2.2 and Section 5, paragraph 3).

4. It's not clear what the expectations were for the UR condition in Experiment 2. Supplementary materials indicated that subjects gaze was monitored but it provides no summary about the gaze in the UR condition. But it seems likely that different subjects would make different choices in controlling their gaze. At the end of the paragraph just before the Conclusions section (page 13), the authors attempt some explanation for results in the UR condition, but I can't say I understand this at all. The ending phrase is "simply because step precision was unnecessary in this experiment". Are you referring to the UR condition or to Experiment 2 in general? The UR condition served as the control for steadiness. Although the question that the reviewer raised is highly relevant, the eye-tracker was used to establish the compliance of our participants, not gaze-behavior generally. Thus, we did not analyze gaze behavior in the UR condition and cannot provide a quantitative answer.

As for the second part of the question; much like the first part of this comment, the second point was made by Reviewer 2 as well. Briefly, in the last part of the discussion we attempt to interpret the results of the walking experiment in the context of a control model proposed by Matthis et al. [19,20], in which it is suggested that gazing downward serves in feedforward to guide locomotion. While this point is not essential to the paper, we believe it can make it more interesting as it extends Matthis' model.

In the original manuscript, we attempted to be as concise as possible. Apparently, this concision came at the cost of vagueness. To make this part clearer (and to address other comments that the reviewers made), we have made extensive changes to last part of the discussion (Section 3.4.2). We hope that these changes clarify the point we were trying to make. However, if the reviewers and/or editor still have concerns about this part, we do not object to removing the last three paragraphs from the manuscript. See below the rewritten section:

As noted earlier, the results of this experiment were not as straight forward as those of the standing experiment and are therefore more difficult to interpret. Nevertheless, a previously published control model, in which gazing downward was suggested to serve in feedforward when stepping onto a specific foothold (i.e., precise stepping) may shed some light. Specifically, these authors suggest that during the single support phase, the body acts as an inverted pendulum, in which the COM rotates around the instantaneous BOS. They argue that once the execution of the step to the target commenced (after push-off), further control and the target itself were not required because the trajectory followed the pendular motion. These authors argue that such feedforward control enables precise stepping, while exploiting the biomechanics of bipedal gait for energetic efficiency. This model suggests a close coupling between the vertical and forward motions, as they are determined by the radius of the pendulum. However, this inverted pendulum model assumes a fixed distance of the COM from the BOS, which is not the case for humans due to knee joint motion. Thus, the radius of the pendulum must be controlled throughout the single support phase to ensure the precision of the swinging leg. Our results suggest that this may be achieved through feedback control, by exploiting the visual flow

created while gazing downward. If so, the effect observed for the Y-axis motion and that of the vertical motion might not be mutually exclusive, because both are influenced by the radius of the pendulum. Since we monitored the ground reaction force and COP, and not the motion of the COM, the magnitudes of these components are not expected to be tightly coupled, but some resemblance is expected (as was observed).

This perspective not only demonstrates the interplay between the visual information used to guide locomotion and that used to control posture, but also provides a reasonable explanation as to why vertical steadiness decreased during the UR condition (in comparison to the DWG1 and DWG3 conditions)—that is, simply because step precision was unnecessary in this experiment. Namely, when free to act as they please, participants did not try to exploit the visual flow to achieve step precision simply because it was redundant, or in other words, steadiness was not perceived as necessary or beneficial to the goals of the task.

5. Figure 1 and Experiment 1 analysis. The Drs measure is calculated by combining X and Y COP measurements so Drs is not independent of Dxs and Dys. Therefore it's a bit misleading to even bother showing and commenting on the Drs results separately since the implication, unless you think about it carefully, is that Drs results are somehow providing some independent confirmation for results seen in the other two measures, but that's not true. My suggestion would be to not include the Drs analysis.

While we agree with the reviewer that mathematically $Drs = Dys + Dxs$, we believe it is the other way around: Drs is the main outcome measure representing the 2D motion of the COP, and the other two coefficients represent its breakdown into its 1D components. Further, we think that the Drs does provide additional information—namely, the comparison between the DWGF and FG conditions revealed significant differences between them for each 1D coefficient, but not the 2D coefficient (Drs), for which no difference was observed.

6. All figures. A good general practice is to 'show the data'. All figures are only showing mean and two times standard errors. This gives a reader no feeling for the distribution of the data. Better would be to show the individual data points along side some summary representation (mean and SE or $2*SE$).

Individual data points are usually shown for small data sets. In the first experiment we had 375 data points and in the second 150 data points. Further, our figures are the only way that the reader can see the pairwise comparisons between conditions. Formatting the figures according to the reviewer's suggestion would prevent us from showing these pairwise comparisons (see an example in Figure 3 below) and would require the addition of one or two tables to the manuscript. Although we agree with the reviewer that showing individual data points gives the reader a better sense of the data and do not object to changing our figures, we feel it is ultimately an editorial choice. If requested to do so, we will.

Figure 3: An example of the appearance of our graphs if formatted according to the reviewer’s suggestion. It is evident from this figure that the pairwise comparisons would be much harder to incorporate within such graph.

Minor comments

1. Page 4, line 35. The general definition of steadiness in Experiment 1 is defined in terms of COM motion but the actual measures used in your analysis are derived from center of pressure (COP) motion which, for inverted pendulum mechanics, is related to the ankle torque exerted to control COM and is a function of COM displacement and COM acceleration.

The text has been revised according to the reviewer’s comment.

2. Page 4, line 51 and page 16 line 14. The authors use the term “displacement coefficient” rather than the original “diffusion” coefficient. It seems unwise to redefine a well-established established term. Additionally, it's a slope measure and not a distance measure.

The text has been revised according to the reviewer’s comment.

3. Figure 1 y-axis labels and page 16, line 33. Scientific units for seconds is ‘s’ not ‘sec’.

The figure and text have been revised according to the reviewer’s comment.

4. Page 6, line 29. Again, you are mentioning COM but your calculations are based on COP and Fz. Also, is referring to the BOS relevant for gait, as compared to stance, since the BOS is changing over time? Also, somewhere (probably Methods section) you should make clear that your COP measure is the whole body COP (I think it is) and not COP measure under each foot. Maybe your treadmill only has one force plate and that’s what you get, but other systems can have separate force plates for each side.

For page 6, line 29. The sentence states that this motion is estimated from the ground reaction force and COP.

For Section 2.2, first paragraph, and Section 6.3, third paragraph. We have revised the text to state that this was the COP of the whole body, as suggested by the reviewer.

5. Page 6, line 50. Does this 5.9 cm difference indicate that the subject was farther from the target or closer to the target?

The text has been revised to clarify that the distance between the walker and the target increased by 5.9 cm.

6. Page 7, line 31. Should be 'values'.

The text has been revised.

7. Pages 9, 10. The discussion of isotropy and anisotropy is related to the visual conditions, but these X-Y differences will also be related to stance width (see some earlier work by Brian Day). I note that your data appears to show anisotropy in the EC condition which obviously would not be due to visual conditions.

In the first experiment we chose to test our participants in a narrow base stance, because we were concerned about a floor effect. A narrow base stance is more challenging and therefore more sensitive to postural control [5] also, compared to a wide base stance, the narrow stance was reported to produce greater sway along the X axis than along the Y axis, in both young adults and elderly people [5,21]. The reason for this difference is most likely related to the biomechanics of the human body and not to visual information. The results of the EC condition confirm that this is the case. Naturally, stance width, as well as other parameters, can affect the ratio between the two directions. We have added an explanation as to why ML sway is expected to be greater in the current investigation. (See Sections 3.1 and 6.2.)

8. Page 16, line 29. The use of the 'i' symbol on this line is confusing. Apparently it's meant of apply to X, Y, or R but 'i' is also used in the above equation.

The text has been revised to exclude the "i" and avoid to confusion.

9. Page 18, paragraph starting on line 18. This is not a very clear description. This explanation would benefit from have a methods figure that shows where your algorithm has picked off points from the COPX and GRF data indicating heel strike times.

Figure 4: A graphical representation of the algorithm that detects the heel-strikes from the vertical ground reaction force and the trajectory of the COPX. The heel strike was determined at the time-point at which the greatest change in the ground reaction force occurred (steepest slope), just after the rapid change in the trajectory of the COPX time series.

10. Page 18, line 38. The Chau et al. reference is not in the reference list. This reference has been added to the reference list.

Reviewer: 2

Comments to the Author(s)

This manuscript investigates the contribution of gazing down towards postural control. Subjects stood and walked while gazing at different locations ranging from straight ahead to down at the feet. These two studies (standing, walking) found that gazing downward a few meters ahead resulted in more steady standing and walking, quantified by stabilogram diffusion analysis and Lyapunov exponents. These results have implications for the use of visual information for locomotion guidance and control.

Main concerns:

The manuscript is generally well-written and detailed. I have some main concerns that might provide additional clarity to the studies and to the implications of the results.

The manuscript was somewhat difficult to read through because of the extensive use of acronyms and X and Y (e.g. X-axis, Dxs, COPX). It was not clear to me initially which directions X and Y referred to. A brief explanation does exist in Results for Experiment 1 (Page 3), but it would be best if it were repeated in the Results section as well. Perhaps it would help the reader to change X and Y to ML and AP, respectively.

Following this suggestion, we have replaced X and Y with ML and AP, respectively, in several places and added a reminder in other places that terms with X relate to the ML and those with Y relate to the AP.

For Experiment 2, it is unclear why walking speed was not kept consistent throughout all the trials for each subject or among all subjects. This decision resulted in analyses that needed to account for velocity to show significance. Some explanation for the rationale behind letting subjects adjust walking speed between gaze conditions would be beneficial and any influence of allowing self-selected speed on the limitations of the experiment.

The choice to allow participants to select a different velocity for each condition was made because any deviation from the preferred velocity, much like velocity itself, can affect walking steadiness [22,23]. If the visual condition affects the choice of velocity, then forcing a certain velocity could lead to a change in performance that is unrelated to the visual flow. We assumed that controlling for walking velocity would be easier than controlling for the deviation from the preferred velocity. A short explanation was added to the text (Section 6.3, paragraph 2).

It is interesting that the UR (unrestricted) condition did not perform the best. The authors posited that perhaps step precision (which I interpreted as control) was unnecessary in this experiment. I am not sure about this explanation. Presumably, control is needed for all conditions to not, for example, fall off the treadmill. Do the authors mean that less control was needed for UR than for the DWG1 and DWG3 condition because it was more natural for the subjects?

See the answer to Reviewer 1 (4th comment).

One of the main conclusions is that downward gazing could be beneficial for postural control. For UR, did subjects prefer to adopt a downward gaze? Was it within one to three meters?

See the answer to reviewer 1 (4th comment).

While I agree the explanation for gaze compliance should be in the appendix, the results section should give some indication that the subjects complied (and therefore the measures from these conditions are valid).

We have added a short qualitative description of these results in the manuscript (Section 6.3, paragraph 1).

Minor comments:

Page 9 Line 24: "However, these previous mechanistic investigations had used gaze distances irrelevant to walking." Irrelevant in what way?

The text has been revised to clarify that these reports included very short eye-target distance (such as 45 cm), which are irrelevant for walking.

Page 10 Line 46: "We believe participants were trying (unconsciously) to optimize a certain parameter..." Please clarify what these certain parameters are.

The text has been revised to clarify that this is only a speculation; we don't know what this parameter is and whether or not it is related to the optical flow.

Page 10 Line 52: Optimal for what? Related to energetics or stability or something else?

The text has been revised to clarify that such gaze behavior is optimal for feedforward control of step precision.

Page 11 Line 48: Do the authors mean "equivalent" or "similar", instead of "equivocal"?

Indeed. The text has been revised.

Reference list

[1] Collins, J. J., & De Luca, C. J. (1993). Open-loop and closed-loop control of posture: a random-walk analysis of center-of-pressure trajectories. *Experimental brain research*, 95(2), 308-318.

[2] Collins JJ, De Luca CJ. The effects of visual input on open-loop and closed-loop postural control mechanisms. *Exp Brain Res*. 1995;103(1):151-63. doi: 10.1007/BF00241972. PMID: 7615030.

- [3] Laughton, C. A., Slavin, M., Katdare, K., Nolan, L., Bean, J. F., Kerrigan, D. C., ... & Collins, J. J. (2003). Aging, muscle activity, and balance control: physiologic changes associated with balance impairment. *Gait & posture*, *18*(2), 101-108.
- [4] Mitchell, S. L., Collin, J. J., De Luca, C. J., Burrows, A., & Lipsitz, L. A. (1995). Open-loop and closed-loop postural control mechanisms in Parkinson's disease: increased mediolateral activity during quiet standing. *Neuroscience letters*, *197*(2), 133-136.
- [5] Melzer, I., Benjuya, N., & Kaplanski, J. (2004). Postural stability in the elderly: a comparison between fallers and non-fallers. *Age and ageing*, *33*(6), 602-607.
- [6] Melzer, I., Kurz, I., & Oddsson, L. I. (2010). A retrospective analysis of balance control parameters in elderly fallers and non-fallers. *Clinical Biomechanics*, *25*(10), 984-988.
- [7] Aoki, O., Otani, Y., Morishita, S., & Domen, K. (2015). EFFECTS OF VIEWING DISTANCE AND HEAD FLEXION ON POSTURAL CONTROL DURING ONE AND TWO-LEGGED STANCE. *Int J Physiother Res*, *3*(5), 1215-20.
- [8] Aoki, O., Otani, Y., Morishita, S., & Domen, K. (2014). Influence of gaze distance and downward gazing on postural sway in hemiplegic stroke patients. *Experimental brain research*, *232*(2), 535-543.
- [9] Dingwell, J. B., & Cusumano, J. P. (2000). Nonlinear time series analysis of normal and pathological human walking. *Chaos: An Interdisciplinary Journal of Nonlinear Science*, *10*(4), 848-863.
- [10] Rosenstein, M. T., Collins, J. J., & De Luca, C. J. (1993). A practical method for calculating largest Lyapunov exponents from small data sets. *Physica D: Nonlinear Phenomena*, *65*(1-2), 117-134.
- [11] Bruijn, S. M., Meijer, O. G., Beek, P. J., & van Dieën, J. H. (2013). Assessing the stability of human locomotion: a review of current measures. *Journal of the Royal Society Interface*, *10*(83), 20120999.
- [12] Stergiou, N., Moraiti, C., Giakas, G., Ristanis, S., & Georgoulis, A. D. (2004). The effect of the walking speed on the stability of the anterior cruciate ligament deficient knee. *Clinical Biomechanics*, *19*(9), 957-963.
- [13] Kang, H. G., & Dingwell, J. B. (2008). Effects of walking speed, strength and range of motion on gait stability in healthy older adults. *Journal of biomechanics*, *41*(14), 2899-2905.
- [14] Lockhart, T. E., & Liu, J. (2008). Differentiating fall-prone and healthy adults using local dynamic stability. *Ergonomics*, *51*(12), 1860-1872.
- [15] Yakhdani, H. R. F., Bafghi, H. A., Meijer, O. G., Bruijn, S. M., van den Dikkenberg, N., Stibbe, A. B., ... & van Dieën, J. H. (2010). Stability and variability of knee kinematics during gait in knee osteoarthritis before and after replacement surgery. *Clinical biomechanics*, *25*(3), 230-236.

- [16] Kao, P. C., Dingwell, J. B., Higginson, J. S., & Binder-Macleod, S. (2014). Dynamic instability during post-stroke hemiparetic walking. *Gait & posture*, *40*(3), 457-463.
- [17] McAndrew, P. M., Wilken, J. M., & Dingwell, J. B. (2011). Dynamic stability of human walking in visually and mechanically destabilizing environments. *Journal of biomechanics*, *44*(4), 644-649.
- [18] Dingwell, J. B., Cusumano, J. P., Cavanagh, P. R., & Sternad, D. (2001). Local dynamic stability versus kinematic variability of continuous overground and treadmill walking. *J. Biomech. Eng.*, *123*(1), 27-32.
- [19] Matthis, J. S., & Fajen, B. R. (2013). Humans exploit the biomechanics of bipedal gait during visually guided walking over complex terrain. *Proceedings of the Royal Society B: Biological Sciences*, *280*(1762), 20130700.
- [20] Matthis, J. S., Barton, S. L., & Fajen, B. R. (2017). The critical phase for visual control of human walking over complex terrain. *Proceedings of the National Academy of Sciences*, *114*(32), E6720-E6729.
- [21] Melzer, I., Benjuya, N., & Kaplanski, J. (2001). Age-related changes of postural control: effect of cognitive tasks. *Gerontology*, *47*(4), 189-194.
- [22] Dingwell, J. B., & Marin, L. C. (2006). Kinematic variability and local dynamic stability of upper body motions when walking at different speeds. *Journal of biomechanics*, *39*(3), 444-452.
- [23] England, S. A., & Granata, K. P. (2007). The influence of gait speed on local dynamic stability of walking. *Gait & posture*, *25*(2), 172-178.

Appendix B

Dear Editor,

We were pleased to hear you decided to accept our manuscript for publication. We wish to thank you again for the time and effort invested in our manuscript. Below you can find our point-by-point response to the additional comments.

Editorial Comments:

We note that there were some issues regarding the figures within your submission. When submitting your finalised revision, please ensure to include your final figures in-text where you wish for them to be placed (along with the appropriate figure caption), **and** as separate figure files for our Production team.

The figures and captions were added.

Reviewer: 2

The amended text about subject compliance states, "and found to be excellent in 62% of the trials, very good in 31%, and fair in 3%." I am unsure how the authors determined or how readers should interpret what is considered excellent, very good, or fair, and there does not seem to be an explanation in the Appendix either.

An explanation was added to the Appendix.

The authors noted that they could not evaluate whether downward gaze was adopted for the UR condition because gaze was only evaluated for subject compliance. It would be beneficial to include that in the manuscript, perhaps in the Appendix. A natural question leading from the results is whether the UR gaze was in the downward direction, and that note would let the readers know that it remains unknown.

We have added to the Appendix a statement to this effect (second paragraph, last sentence).

Supplementary files containing data are in .sav and .sps, which I cannot open. I am not sure if supporting data is included with these SPSS files. The data files should be included in a format that is accessible by most software programs (e.g. in .csv or .txt).

SPSS files were replaced with .csv and .txt files.

Reviewer: 1

Comments to the Author(s)

Major comment

1. Page 14, line 37. The manuscript states that the diffusion coefficient was calculated as the slope of the $\Delta_{\text{distance_squared}}$ by Δ_{t} plot. But the 1993 Collins and De Luca paper defines the diffusion coefficient as one half the slope (equation 1 and Figure 2b in Collins and De Luca; e.g. slope = $2 * D_{rs}$). If the authors actually are reporting the slope and not one half the slope, then Figure 1 and the description of the D calculation needs to be corrected. If the authors did the diffusion coefficient calculation correctly (according to Collins and De Luca) then the description of the calculation on page 14 needs to be corrected.

The diffusion coefficient was calculated as described by Collins and De-Luca, i.e., $0.5 * \text{slope}$. We have revised the text to clarify (section 6.2, 5th paragraph).

Minor comments

1. Page 3, line 45/46. This sentence mentions “a standardized, narrow-base stance”. Exactly what the foot placement is for narrow-base stance should be defined in the Methods section. Were feet fully together or was there some space between the feet?

The feet were tight together as stated in section 6.2, second paragraph: “Participants were instructed to stand barefoot....in a standardized stance, i.e. with their feet tight together and hands loosely hanging at their sides.”

2. Page 5, line 46. A couples of spaces are needed in this parenthetic F statistic description.

These were added.

3. Page 15, line 12. This sentence mentions how the results were “similar” to results from the SDA analysis when more traditional parameters were used. This is a bit of a weak statement. Can the authors make a stronger claim? Maybe something like “essentially identical such that overall conclusions were unchanged” (assuming this is true – if not true then some more details are necessary).

The sentence was rephrased to state that results were “essentially identical.”

4. Page 15, lines 47-49. “ETG” is not defined yet. I see in the Appendix that this apparently stands for eye tracking device. It is not worth defining a new acronym here.

The acronym was replaced with “eye-tracking glasses.”

5. Page 15, first paragraph in section 6.3. Various percentages are given for how well subjects were able to comply with the gaze directions. It seems that there should be some correspondence between these percentages and those reported in the Appendix, but there doesn’t seem to be.

An explanation was added to the Appendix.

6. Page 16, line 50. This description for identifying the heel strike point is not clear. First, as currently written the phrase “exceeding body weight” seems to be saying something about the “change in COPX trajectory” but apparently it should apply to the first part of the sentence. That is, “. . . a surge in GRF, exceeding body weight, followed the onset of a rapid change in COPX trajectory.” Then the following sentence is also not clear given the figure that the authors included in their response to the reviewers. As stated, the heel strike would be identified at the peak of the GRF since this is the point where there is a “maximal increase in force”. Maybe say “The heel-strike was defined as the point where the COPX begins its rapid change and the GRF begins its surge.” Or something similar.

The sentences were rephrased according to the reviewer’s suggestion.

7. I note that reviewer #2 wanted to see that ML and AP labeling was used rather than, or in addition to, the X and Y labels. This should also apply to the figure legends and, better yet, the headings above the panels in Figures 1 and 3 should have the ML and AP labels.

These labels were added to the figure’s captions.